# No-Regret Thompson Sampling for Finite-Horizon Markov Decision Processes with Gaussian Processes

**Jasmine Bayrooti**
University of Cambridge
jgb52@cam.ac.uk

**Sattar Vakili**
MediaTek Research
sattar.vakili@mtkresearch.com

**Amanda Prorok**
University of Cambridge
asp45@cam.ac.uk

**Carl Henrik Ek**
University of Cambridge
Karolinska Institutet
che29@cam.ac.uk

## Abstract

Thompson sampling (TS) is a powerful and widely used strategy for sequential decision-making, with applications ranging from Bayesian optimization to reinforcement learning (RL). Despite its success, the theoretical foundations of TS remain limited, particularly in settings with complex temporal structure such as RL. We address this gap by establishing no-regret guarantees for TS using models with Gaussian marginal distributions. Specifically, we consider TS in episodic RL with joint Gaussian process (GP) priors over rewards and transitions. We prove a regret bound of $\tilde{\mathcal{O}}(\sqrt{KH\Gamma(KH)})$ over $K$ episodes of horizon $H$, where $\Gamma(\cdot)$ captures the complexity of the GP model. Our analysis addresses several challenges, including the non-Gaussian nature of value functions and the recursive structure of Bellman updates, and extends classical tools such as the elliptical potential lemma to multi-output settings. This work advances the understanding of TS in RL and highlights how structural assumptions and model uncertainty shape its performance in finite-horizon Markov Decision Processes.

## 1 Introduction

Sequential decision-making under uncertainty lies at the core of many machine learning systems, from robotics (Kober et al., 2013) and chip design (Mirhoseini et al., 2021) to large language models (Ouyang et al., 2022). In these settings, an agent must make a series of decisions, balancing exploration to learn about the environment with exploitation to act effectively. A central question is: how should an agent leverage uncertainty to act optimally over time?

Thompson sampling (TS) is a widely used and principled approach for sequential decision-making that naturally balances exploration and exploitation through posterior sampling (Thompson, 1933). The idea is to sample a model from the posterior and select actions that are optimal under that model. The inherent randomness in TS aligns exploration with the agent's uncertainty and can lead to more efficient learning compared to methods based on confidence bounds (Russo et al., 2018). TS underpins a range of applications including bandits (Chapelle and Li, 2011; Kaufmann et al., 2012b; Russo et al., 2018), Bayesian optimization (Srinivas et al., 2010; Chowdhury and Gopalan, 2017; Vakili et al., 2021b), and reinforcement learning (RL) (Sasso et al., 2023; Osband and Van Roy, 2017; Bayrooti et al., 2025). Empirically, TS has demonstrated strong performance across domains and is widely adopted in practice. Theoretical guarantees for TS have been well-developed in multi-armed bandit settings (Agrawal and Goyal, 2012; Kaufmann et al., 2012a) and have also been extended to RL settings (Osband et al., 2013; Osband and Van Roy, 2017; Chowdhury and Gopalan, 2019; Dann

39th Conference on Neural Information Processing Systems (NeurIPS 2025).

et al., 2021). However, existing analyses in RL typically rely on discrete state-action spaces, assume linear or kernelized dynamics, or yield regret bounds that scale poorly with state dimensionality. Establishing tight regret bounds for TS in continuous-state Markov Decision Processes (MDPs) without strong structural assumptions remains an important open problem.

In this work, we take a step toward closing this gap by studying TS for sequential decision-making under general, continuous models of the environment. We focus on RL in finite-horizon MDPs, where both rewards and transitions are jointly modeled using a multi-output GP, as proposed by Bayrooti et al. (2025). This approach enables the agent to model correlations across different components of the environment in a flexible and data-efficient manner. We consider an episodic MDP with $K$ episodes of horizon $H$, where at the start of each episode, the agent samples a realization from the GP posterior and computes an optimal policy with respect to this sample. Regret is defined as the cumulative loss in value relative to the optimal policy. We provide a stylized analytical upper bound on the regret, showing its dependence on the number of episodes, the horizon, and the complexity of the GP kernel. Our analysis highlights the key theoretical challenges of applying TS in sequential settings, particularly due to the recursive and compositional nature of value functions in RL.

## 1.1 Contributions

We establish a sublinear regret bound for TS in model-based RL under GP models, an approach we refer to as *Reinforcement Learning with GP Sampling (RL-GPS)*. We prove a regret bound of $\tilde{\mathcal{O}}(\sqrt{KH\Gamma(KH)})$ over $K$ episodes of horizon $H$, where $\Gamma(\cdot)$ captures the complexity of the GP model. Sublinear regret in $K$ implies that RL-GPS asymptotically matches the performance of the optimal policy, referred to as *no-regret* learning (Srinivas et al., 2010; Agrawal and Goyal, 2013b).

Our theoretical analysis introduces several novel intermediate contributions that are essential to deriving the final regret bound. Extending TS regret guarantees to RL presents two main challenges: (i) the optimal value function is a recursive composition of GPs which is not a GP itself (Damianou and Lawrence, 2013); and (ii) TS operates on proxy models induced by Bellman updates, rather than sampling directly from the posterior over the optimal value function. To address (i), we derive high-probability confidence bounds for compositional functions of GPs, formalized in Theorem 1, which allow us to control the error that accumulates through Bellman recursion. Building on these bounds, we obtain high-probability confidence intervals for recursive value functions in episodic MDPs (Corollary 1) that links the sampled proxy to the true value. To address (ii), we bound the regret in terms of cumulative posterior uncertainty via a new multi-output *elliptical potential lemma* (Lemma 1) that jointly tracks correlated uncertainty across multiple outputs. This lemma provides tighter regret guarantees than naively applying the standard versions (Srinivas et al., 2010; Abbasi-Yadkori et al., 2011; Carpentier et al., 2020) independently to each output dimension by leveraging the correlation structure. We further introduce a delayed-update lemma (Lemma 2) that accounts for model updates at the end of episodes, yielding an improved dependence on the horizon $H$.

Finally, we conduct controlled experiments designed to mirror the theoretical assumptions and validate our regret bounds. Our results confirm sublinear cumulative regret across a range of environments, including GP-sampled MDPs and sparse navigation tasks. Furthermore, we empirically illustrate how the choice of GP kernel affects learning efficiency, with smoother kernels such as RBF leading to faster regret decay in smooth environments, and rougher Matérn kernels outperforming in sparse settings. These findings are consistent with the theoretical dependence on model complexity and highlight the importance of model selection in practical applications.

## 1.2 Related work

**RL with TS.** The theoretical performance of Thompson sampling (TS) has been extensively studied in the bandit setting, where it achieves near-optimal regret bounds and strong empirical performance (Agrawal and Goyal, 2012, 2013a; Kaufmann et al., 2012b; Korda et al., 2013; Russo and Van Roy, 2014; Russo et al., 2018; Kveton et al., 2020). In the GP bandit setting, Chowdhury and Gopalan (2017) provided regret bounds for TS under kernel-based assumptions on the target functions, introducing techniques that our analysis builds on (see Step 2 in Section 4 for details). Extending TS to RL introduces new challenges due to the recursive structure of value functions and the dependence on both states and actions. Several works have established foundational regret guarantees for TS in finite MDPs (Osband et al., 2013; Osband and Van Roy, 2014; Osband et al.,

2016; Russo, 2019). In particular, Posterior Sampling for Reinforcement Learning (PSRL) (Osband et al., 2013; Osband and Van Roy, 2017) achieves $\tilde{\mathcal{O}}(H\sqrt{SAT})$ Bayesian regret, where $H$ is the horizon, $S$ the number of states, $A$ the number of actions, and $T$ the total number of steps. Building on their analyses, Fan et al. (2018) studied TS in continuous MDPs and established Bayesian regret bounds. Dann et al. (2021) also considered continuous MDPs, but under linearity assumptions, and derived tighter regret guarantees although their bounds scale poorly with state dimensionality. In the kernelized RL setting, Chowdhury and Gopalan (2019) analyzed TS where the transition probability distribution is assumed to be a fixed function in a reproducing kernel Hilbert space (RKHS) and derived regret bounds that depend on the maximum information gain of the kernel.

Our work differs fundamentally from these approaches in both modeling and analysis. We model the reward and transition functions jointly as a multi-output GP and our analysis provides high-probability bounds that hold uniformly across all problem instances, unlike Bayesian regret bounds that are averaged over the distribution of all problems. Consequently, our approach requires different proof techniques and much of the analysis we present is novel, including Theorem 1 on confidence bounds for composed GPs, Corollary 1 on high-probability value function bounds, and Lemmas 1 and 2 on multi-output elliptical potential bounds. Our assumptions are also more mild than in prior analyses. For example, PSRL (Osband et al., 2013; Osband and Van Roy, 2017) assumes finite state and action spaces, linear MDP methods (Dann et al., 2021; Jin et al., 2020) assume linearity in both rewards and transitions, and RKHS-based approaches (Chowdhury and Gopalan, 2019) assume that the transition dynamics are modeled as fixed functions within a known RKHS, effectively treating each component independently. In contrast, our multi-output GP framework flexibly captures correlations between reward and transition without requiring assumptions on discretization, linearity, or independence.

**Episodic MDP.** The episodic MDP framework is a central setting for RL. Sublinear regret bounds have been established for Upper Confidence Bound (UCB) based methods in tabular finite-horizon MDPs (Azar et al., 2017; Jin et al., 2018; Zanette and Brunskill, 2019). Subsequent works have developed regret analyses for UCB-style methods with additional structural assumptions, including linear (Jin et al., 2020) and kernelized MDPs (Chowdhury and Gopalan, 2019; Yang and Wang, 2020; Vakili and Olkhovskaya, 2023). Additionally, Curi et al. (2020) introduced an optimistic algorithm for continuous state-action MDPs and established regret guarantees under standard assumptions for GP models. These approaches all rely on optimism via constructed confidence sets to guide exploration. In contrast, TS offers an exploration approach that avoids explicit construction of confidence sets and has been less theoretically studied in complex RL settings.

**Broader RL settings.** Beyond episodic MDPs, it is also common to study performance in infinite-horizon discounted (Puterman, 2014) and average-reward settings (Auer et al., 2008; Vakili and Olkhovskaya, 2024). In the discounted setting, the contraction properties of the Bellman operator enable efficient learning. For example, Ouyang et al. (2017) leverages these properties to dynamically adapt episode lengths and demonstrate sublinear regret for TS. In the infinite-horizon average-reward setting, regret bounds have been established for tabular communicating MDPs with finite diameter using UCB-based strategies (Auer et al., 2008; Jaksch et al., 2010; Agrawal and Jia, 2017; Wei et al., 2021). More recent works have analyzed regret in structured infinite-horizon settings, each addressing different challenges. Wu et al. (2022) provided nearly minimax-optimal guarantees for model-based linear mixture MDPs with known feature mappings. Ghosh and Zhou (2023) established sublinear regret bounds using model-free methods with linear function approximation in weakly communicating MDPs. Sukhija et al. (2024) studied continuous, nonlinear dynamical systems in nonepisodic settings and established sublinear regret bounds in terms of maximum information gain under continuity and bounded energy assumptions. In general, infinite-horizon and average-reward settings require structural assumptions such as finite diameter, weakly communicating structure, or continuity and boundedness to obtain theoretical guarantees and the resulting regret bounds often depend explicitly on these properties. By focusing on the episodic setting, we sidestep these complications and exploit the finite-horizon structure to develop a regret analysis using GP-based recursive bounds.

## 2 Problem formulation

An episodic MDP is defined by the tuple $(\mathcal{S}, \mathcal{A}, f_{\mathrm{R}}, f_{\mathrm{S}}, H)$, where $\mathcal{S} \subset \mathbb{R}^{d_{\mathrm{S}}}$ is the state space, $\mathcal{A} \subset \mathbb{R}^{d_{\mathrm{A}}}$ is the action space, and $H$ is the episode length. The reward function is $f_{\mathrm{R}} : \mathcal{S} \times \mathcal{A} \mapsto \mathbb{R}$, and the state transition function is $f_{\mathrm{S}} : \mathcal{S} \times \mathcal{A} \mapsto \mathcal{S}$.

The policy $\pi = \{\pi_h : \mathcal{S} \mapsto \mathcal{A}\}_{h=1}^H$ specifies the action $\pi_h(s)$ the agent takes in state $s$ at step $h$. At the start of each episode $k = 1, 2, \ldots, K$, the environment selects an initial state $s_{1,k}$ and the agent determines a policy $\pi_k = \{\pi_{h,k}\}_{h=1}^H$. At each step $h$ of the episode, the agent observes the state $s_{h,k}$ and selects action $a_{h,k} = \pi_{h,k}(s_{h,k})$. The agent then receives reward $f_\mathrm{R}(s_{h,k}, a_{h,k})$ and transitions to the new state $s_{h+1,k} = f_\mathrm{S}(s_{h,k}, a_{h,k})$.

In an episodic MDP, the agent aims to maximize the cumulative reward collected over an episode. To formalize this, we define the *value function* of a policy $\pi$ as the expected total reward obtained when starting at state $s$ at step $h$ and following $\pi$ thereafter, where the expectation is taken over the trajectory $\{(s_{h'}, a_{h'})\}_{h'=h}^H$ induced by the policy $\pi$:

$$V_h^\pi(s) = \mathbb{E}\left[\sum_{h'=h}^H f_\mathrm{R}(s_{h'}, a_{h'}) \mid s_h = s\right], \quad \forall s \in \mathcal{S}, \ h \in [H]. \tag{1}$$

The associated *state-action value function* is defined as:

$$Q_h^\pi(s, a) = \mathbb{E}\left[\sum_{h'=h}^H f_\mathrm{R}(s_{h'}, a_{h'}) \mid s_h = s, a_h = a\right]. \tag{2}$$

We assume the existence of an optimal policy $\pi^\star$ that maximizes the expected total reward from any state and time step. The optimal value and optimal state-action value functions are defined as:

$$V_h^\star(s) = \max_\pi V_h^\pi(s), \quad Q_h^\star(s, a) = \max_\pi Q_h^\pi(s, a). \tag{3}$$

The optimal value function satisfies the Bellman optimality equation:

$$Q_h^\star(s, a) = f_\mathrm{R}(s, a) + V_{h+1}^\star(f_\mathrm{S}(s, a)), \quad V_h^\star(s) = \max_{a \in \mathcal{A}} Q_h^\star(s, a), \tag{4}$$

with $V_{H+1}^\star(s) = 0$ for all $s \in \mathcal{S}$. An RL algorithm aims to find a near-optimal policy while interacting with the environment. The *regret* over $T$ timesteps is defined as:

$$\mathrm{Regret}(T) = \sum_{k=1}^K (V_1^\star(s_{1,k}) - V_1^{\pi_k}(s_{1,k})), \tag{5}$$

where $\pi_k$ is the policy executed by the agent in episode $k$, and $s_{1,k}$ is the initial state of that episode, and we use $T = KH$ for the total number of steps.

**Gaussian process modeling.** GPs specify distributions over the space of functions, offering calibrated uncertainty estimates that can be leveraged for exploration and decision making. In the single-output case, we model an unknown function $f : \mathcal{Z} \to \mathbb{R}$ as a Gaussian process:

$$f \sim \mathrm{GP}(0, k), \tag{6}$$

with a scalar-valued kernel $k : \mathcal{Z} \times \mathcal{Z} \to \mathbb{R}$. Given $n$ noisy observations $\{(z_i, y_i)\}_{i=1}^n$ with $y_i = f(z_i) + \varepsilon_i$ and $\varepsilon_i \sim \mathcal{N}(0, \lambda^2)$, the posterior mean and variance at any test point $z \in \mathcal{Z}$ are given by:

$$\mu_n(z) = \mathbf{k}_n^\top (\mathbf{K}_n + \lambda^2 \mathbf{I}_n)^{-1} \mathbf{y}_n,$$
$$\sigma_n^2(z) = k(z, z) - \mathbf{k}_n^\top (\mathbf{K}_n + \lambda^2 \mathbf{I}_n)^{-1} \mathbf{k}_n, \tag{7}$$

where $\mathbf{K}_n \in \mathbb{R}^{n \times n}$ is the kernel matrix with $[\mathbf{K}_n]_{ij} = k(z_i, z_j)$, $\mathbf{k}_n(z) \in \mathbb{R}^n$ has entries $k(z_i, z)$, and $\mathbf{y}_n \in \mathbb{R}^n$ is the vector of observed outputs.

**Multi-output Gaussian processes.** In many applications, we wish to model a vector-valued function $f : \mathcal{Z} \to \mathbb{R}^d$ jointly across multiple correlated outputs. In this setting, $f$ is modeled as a multi-output GP (6) where $k : \mathcal{Z} \times \mathcal{Z} \to \mathbb{R}^{d \times d}$ is a matrix-valued positive semidefinite kernel that encodes both input similarity and output correlations.

Given $n$ input points $z_1, \ldots, z_n$, let the observed outputs be collected into a vector $\mathbf{y}_n \in \mathbb{R}^{nd}$ by stacking all $d$ outputs at each input. The full joint prior over $\mathbf{y}_n$ is a multivariate Gaussian with zero mean and block kernel matrix $\mathbf{K}_n \in \mathbb{R}^{nd \times nd}$ defined by:

$$\mathbf{K}_n[(i-1)d + r, (j-1)d + s] = [k(z_i, z_j)]_{rs} \quad \text{for } i, j \in [n], \ r, s \in [d].$$

The posterior at test point $z$ is again Gaussian:

$$\mu_n(z) = \mathbf{k}_n(z)^\top (\mathbf{K}_n + \lambda^2 \mathbf{I}_{nd})^{-1} \mathbf{y}_n \in \mathbb{R}^d,$$
$$\Sigma_n(z) = k(z,z) - \mathbf{k}_n(z)^\top (\mathbf{K}_n + \lambda^2 \mathbf{I}_{nd})^{-1} \mathbf{k}_n(z) \in \mathbb{R}^{d \times d}, \tag{8}$$

where $\mathbf{k}_n(z) \in \mathbb{R}^{nd \times d}$ is the cross-covariance between $f(z)$ and the training outputs, defined via $[\mathbf{k}_n(z)]_{(i-1)d+r,s} = [k(z_i, z)]_{rs}$. We define $\sigma_n^2(z) := \mathrm{diag}(\Sigma_n(z)) \in \mathbb{R}^d$ as the marginal predictive variances. With slight abuse of notation, we use $\sigma_n(z)$ to denote the vector of marginal standard deviations, where $\sigma_{n,i}(z) = (\sigma_{n,i}^2(z))^{1/2}$ for $i = 1, \ldots, d$. The posterior mean $\mu_n(z)$ and uncertainty $\Sigma_n(z)$ allow multi-output GPs to provide joint, uncertainty-aware predictions across outputs, making them well-suited for RL settings where transition and reward models must be estimated simultaneously.

## 3 Reinforcement learning with GP sampling

In this section, we present RL-GPS for learning episodic MDPs with joint GP modeling of the reward and transition functions, following the multi-output model in Bayrooti et al. (2025).

**Assumption 1.** *Let $f = [f_R, f_S]$ denote the joint reward and transition function. We assume $f$ is distributed as a multi-output Gaussian process: $f \sim GP(\mathbf{0}, k)$, for a known matrix-valued kernel $k : \mathcal{Z} \times \mathcal{Z} \mapsto \mathbb{R}^{d \times d}$.*

**Remark 1.** *GPs offer flexible representational capacity since their smoothness and expressiveness depend on the kernel choice. For instance, the Matérn family introduces a smoothness parameter $\nu$ controlling function regularity, where smaller $\nu$ values yield rougher functions that better capture non-smooth behavior. As shown by Srinivas et al. (2010), Matérn kernels can approximate any continuous function on compact subsets of $\mathbb{R}^d$, making GP priors highly expressive. Consequently, our setting can capture a broad class of reward and transition functions, including those with limited smoothness, and is applicable to many continuous-control environments (Bayrooti et al., 2025).*

The RL-GPS algorithm follows a value-iteration-based form of TS, where at the start of each episode, the agent samples a realization of the reward and transition functions from the GP posterior and computes proxy value functions $Q_{h,k} : \mathcal{Z} \mapsto \mathbb{R}$ via backward induction. The agent then executes the greedy policy induced by this value function for the duration of the episode. This sampling-based approach encourages exploration by introducing structured randomness into value estimates, naturally balancing exploitation of high-reward regions with exploration of uncertain areas of the state-action space. Pseudocode is provided in Algorithm 1.

---

**Algorithm 1** RL with GP Sampling (RL-GPS)

---

1: **Require:** number of episodes $K$, episode length $H$, GP kernel $k$
2: **Initialize:** reward-dynamics model $p(f)$, reward and transition buffer $\mathcal{D}$
3: **for** episode $k = 1, \ldots, K$ **do**
4:      // Create the proxy value functions $Q_{h,k}$
5:      Sample functions $[\hat{f}_{R,k}, \hat{f}_{S,k}]$ from GP posterior $p(f \mid \mathcal{D})$
6:      Initialize $V_{H+1,k}(\cdot) = 0$
7:      **for** $h = H, \ldots, 1$ **do**
8:          $Q_{h,k}(s,a) = \hat{f}_{R,k}(s,a) + V_{h+1}(\hat{f}_{S,k}(s,a))$
9:          $V_{h,k}(s) = \max_{a \in \mathcal{A}} Q_{h,k}(s,a)$
10:     // Follow the greedy policy with respect to $Q_{h,k}$
11:     Observe initial state $s_{1,k}$
12:     **for** $h = 1, \ldots, H$ **do**
13:          Select action $a_{h,k} = \mathrm{argmax}_{a \in \mathcal{A}} Q_{h,k}(s_{h,k}, a)$
14:          Observe next state $s_{h+1,k} = f_S(s_{h,k}, a_{h,k})$ and reward $r_{h,k} = f_R(s_{h,k}, a_{h,k})$
15:          Store reward and transition in buffer $(s_{h,k}, a_{h,k}, r_{h,k}, s_{h+1,k}) \to \mathcal{D}$
16:     Update GP posterior $p(f \mid \mathcal{D})$ using new transitions

---

# 4  Analysis

In this section, we derive a regret bound for RL-GPS (Algorithm 1) in episodic MDPs with a multi-output GP model. We introduce intermediate results and provide the regret bound in Theorem 2.

## 4.1  Confidence intervals

To analyze the regret, we require high-probability bounds on the accuracy of GP predictions. For a single-output GP $f$ with posterior mean $\mu_n$ and standard deviation $\sigma_n$, the tail decay of the Gaussian distribution implies that, with probability at least $1 - \delta$, the following holds uniformly over $z$:

$$|f(z) - \mu_n(z)| \le \beta_n(\delta)\sigma_n(z) \tag{9}$$

where $\beta_n(\delta) = \mathcal{O}(\sqrt{\log(\frac{n}{\delta})})$ (e.g., see Srinivas et al., 2010).

In the RL setting, we must also construct confidence intervals for $v(f_S(\cdot))$ as part of the policy's recursive design where $v : \mathcal{S} \mapsto \mathbb{R}$ is a generic value function. The following theorem addresses this.

**Theorem 1** (Confidence bounds for composed GPs). *Assume $f : \mathcal{Z} \mapsto \mathcal{S} \subset \mathbb{R}^{d_S}$ is a multi-output GP with posterior mean $\mu_n$ and standard deviation $\sigma_n$. Let $v : \mathcal{S} \mapsto \mathbb{R}$ be a twice differentiable value function where for all $s \in \mathcal{S}$, $\|\nabla v(s)\| \le u_G$ and $\|\nabla^2 v(s)\|_{op} \le u_H$. Define the composition $g(z) = v(f(z))$. Then, with probability $1 - \delta$, for all $z \in \mathcal{Z}$,*

$$|g(z) - v(\mu_n(z))| \le u_G \beta_n(\delta/d_S)\|\sigma_n(z)\| + \frac{1}{2}u_H \beta_n(\delta/d_S)^2\|\sigma_n(z)\|^2. \tag{10}$$

The proof uses a Taylor expansion of $v$ to bound $|g(z) - v(\mu_n(z))|$ in terms of the first- and second-order behavior of $v$, together with the standard GP confidence intervals given in (9). A detailed proof is provided in Appendix A.

## 4.2  Performance analysis of RL-GPS

To analyze the regret of RL-GPS, we first introduce our assumption regarding the smoothness of the value functions.

**Assumption 2** (Smoothness of the value functions). *We assume that for all $h$, $V_h$ is twice differentiable where for all $s$, $\|\nabla V_h(s)\| \le u_G$ and $\|\nabla^2 V_h(s)\|_{op} \le u_H$.*

**Remark 2.** *The assumptions on the gradient and Hessian norms are mild compared to those typically imposed on value functions in the literature. For example, in Jin et al. (2020) and Yang et al. (2020), it is assumed that all proxy value functions belong to a function class defined using linear or kernel-based models, respectively. In contrast, we impose a weaker assumption only on the first and second derivatives of the value functions.*

Now we present the main theorem bounding the regret of RL-GPS.

**Theorem 2.** *Consider the episodic MDP setting described in Section 2 and the RL-GPS algorithm given in Algorithm 1. Under Assumptions 1 and 2, with probability $1 - \delta$,*

$$Regret(T) = \mathcal{O}\left(\log(Td/\delta)\sqrt{T\Gamma(T)}\right),$$

*where $\Gamma(T) = \sup_{z_{h,k}, h \in [H], k \in [K]} \mathcal{I}_T$, $\mathcal{I}_T := \frac{1}{2}\log\det(\mathbf{I}_{Td} + \frac{1}{\lambda^2}\mathbf{K}_T)$.*

The determinant in $\Gamma(\cdot)$ represents the complexity of the function space described by the GP (Rasmussen and Williams, 2006) and serves as an upper bound on the information gain, which is discussed in detail in the next section.

The regret bound holds with high probability, where the randomness accounts for both the joint GP distribution of the environment and the randomness in the Thompson samples.

**Remark 3.** *The Matérn family is both theoretically significant and practically prevalent among kernel choices. By substituting the bounds on $\Gamma(\cdot)$ from Vakili et al. (2021a), we obtain the following regret rates. For a base Matérn kernel with smoothness parameter $\nu > 1$, our regret bound becomes:*

$$Regret(T) = \tilde{\mathcal{O}}\left(T^{\frac{\nu+d}{2\nu+d}}\right),$$

*while for the radial basis function (RBF) kernel, the regret bound simplifies to $\tilde{\mathcal{O}}\left(\sqrt{T}\right)$.*

**Remark 4.** *When $H = 1$, learning in episodic MDPs reduces to the degenerate special case of Bayesian optimization, also known as GP bandits. In this setting, we have $T = K$ and our regret bound becomes:*

$$\text{Regret}(T) = \mathcal{O}\left(\log(T/\delta)\sqrt{T\Gamma(T)}\right),$$

*which recovers the standard regret bounds in Bayesian optimization (e.g., see Srinivas et al., 2010).*

### 4.3 Proof of Theorem 2

Our analysis bounds the total regret by comparing the per-episode value of RL-GPS to the optimal value function. We decompose this difference into immediate and recursive components, bound each using confidence intervals derived from the GP model, and then accumulate the bounds via the information gain of the kernel. The key novelty lies in extending both the confidence analysis and the elliptical potential lemma to multi-output GPs. A detailed proof is provided in Appendix B, structured around the following four main steps:

**Step 1: Regret decomposition.** We decompose the per-step regret into two components: an immediate regret term related to TS, and a recursive term capturing uncertainty in value propagation through the transition model (proven in (13) in Appendix B).

$$V_h^\star(s_{h,k}) - V_h^{\pi_k}(s_{h,k}) = \underbrace{Q_h^\star(s_{h,k}, a_{h,k}^\star) - Q_h^\star(s_{h,k}, a_{h,k})}_{\text{(Immediate regret)}} + \underbrace{\left(V_{h+1}^\star(s_{h+1,k}) - V_{h+1}^{\pi_k}(s_{h+1,k})\right)}_{\text{(Recursive part of regret)}}.$$

$$(11)$$

**Step 2: Bounding the immediate regret.** To bound the immediate regret, we build on techniques from the analysis of TS in GP bandits (Chowdhury and Gopalan, 2017), but face two key challenges unique to the RL setting: (i) the target function $Q_h^\star(s, a) = f_R(s, a) + V_{h+1}^\star(f_S(s, a))$ is recursive and compositional and thus cannot be directly modeled as a GP; (ii) the TS algorithm samples from the posterior of a proxy $Q_{h,k}$, not the true posterior over $Q_h^\star$.

To address both issues, we construct high-probability upper and lower confidence bounds on the proxy and target value functions, which to our knowledge have not appeared in prior analyses. These confidence intervals allow us to relate the sampled proxy values to the true value of the selected action and quantify the regret incurred.

**Definition 1** (Upper and lower confidence bounds for value functions)**.** *We define the upper confidence bounds recursively as:*

$$Q_{h,k}^u(s, a) = \mu_{R,k}(s, a) + V_{h+1,k}^u(\mu_{S,k}(s, a)) + \xi_k(s, a), \quad V_{h,k}^u(s) = \max_{a \in \mathcal{A}} Q_{h,k}^u(s, a),$$

*with $V_{H+1,k}^u(s) = 0$. Similarly, the lower confidence bounds are defined as:*

$$Q_{h,k}^l(s, a) = \mu_{R,k}(s, a) + V_{h+1,k}^l(\mu_{S,k}(s, a)) - \xi_k(s, a), \quad V_{h,k}^l(s) = \max_{a \in \mathcal{A}} Q_{h,k}^l(s, a),$$

*with $V_{H+1,k}^l(s) = 0$. The confidence width $\xi_k(s, a)$ is given by:*

$$\xi_k(s, a) = \beta_k(\delta/Td)\sigma_{R,k}(s, a) + u_G\beta_k(\delta/Td)\|\sigma_{S,k}(s, a)\| + \frac{1}{2}u_H\beta_k(\delta/Td)^2\|\sigma_{S,k}(s, a)\|^2,$$

*based on Theorem 1. Here, $\mu_{R,k}$, $\sigma_{R,k}$ and $\mu_{S,k}$, $\sigma_{S,k}$ are the posterior mean and standard deviation of $f_R$ and $f_S$, respectively.*

**Corollary 1.** *With probability at least $1 - \delta$, the optimal and proxy value functions lie within the confidence bounds:*

$$Q_{h,k}^l(s, a) \leq Q_{h,k}(s, a), \ Q_h^\star(s, a) \leq Q_{h,k}^u(s, a), \quad \forall (h, k, s, a).$$

**Step 3: Accumulating regret across an episode.** We unroll the recursive regret decomposition and accumulate the bounds over all steps $h = 1, \ldots, H$ within an episode, leading to a per-episode regret bound expressed in terms of confidence widths:

$$\text{Regret}(T) \leq c\sum_{k=1}^{K}\sum_{h=1}^{H}\xi_k(s_{h,k}, a_{h,k}).$$

where $c$ is an absolute constant.

**Step 4: Bounding cumulative regret.** Finally, we sum the per-episode regret bounds over $K$ episodes, obtaining a cumulative regret expression that involves sums of posterior standard deviations in GPs. We bound these terms using a new elliptical potential lemma (Lemma 1) for multi-output GPs, which is introduced in the next section. Also, an additional challenge arises due to the structure of the problem: it involves a double sum over episodes $k$ and steps $h$, but within each episode, the observations across $h$ are not sequentially incorporated into the GP posterior. This batched structure leads to an additional dependence on the episode length $H$, which we address using tools developed for GP analysis under batch observations and delayed feedback (Calandriello et al., 2020, 2022).

### 4.4 Elliptical potential lemma for multi-output GPs

In analyzing cumulative regret, a central quantity is the sum of sequentially conditioned posterior variances from a GP. A classical result by Srinivas et al. (2010) shows that for scalar-output GPs, the sum of variances along a sequence of inputs $z_1, \ldots, z_T$ is bounded by a log-determinant term:

$$\sum_{t=1}^{T} \sigma_{t-1}^2(z_t) \leq C \log \det \left( \mathbf{I}_{Td} + \frac{1}{\lambda^2} \mathbf{K}_T \right),$$

where $\lambda$ is the GP noise parameter, $\mathbf{K}_T$ is the kernel matrix over the inputs, and $C$ is a constant. This result is often referred to as the *elliptical potential lemma*, particularly in the special case of linear kernels (Carpentier et al., 2020).

In our setting, the transition function is modeled as a multi-output GP, which requires a generalization of the classical result to vector-valued functions. Applying scalar GP bounds independently across output dimensions would result in regret bounds that scale linearly with the state dimension $d_{\mathrm{S}}$. To address this, we derive the following elliptical potential lemma for multi-output GPs that exploits inter-dimensional correlations captured by the kernel.

**Lemma 1** (Elliptical potential lemma for multi-output GPs). *Let $f : \mathcal{Z} \to \mathbb{R}^d$ be a $d$-dimensional function modeled by a multi-output GP with matrix-valued kernel $k$. Let $z_1, \ldots, z_T \in \mathcal{Z}$ be a sequence of input points, and denote by $\sigma_{t-1}(z_t) \in \mathbb{R}^d$ the vector of posterior standard deviations at $z_t$ conditioned on data up to round $t-1$. Then,*

$$\sum_{t=1}^{T} \|\sigma_{t-1}(z_t)\|^2 \leq C \mathcal{I}_T,$$

*where $C = \frac{2}{\log(1+\lambda^{-2})}$ is a constant and $\mathcal{I}_T$ denotes the mutual information between the observations $y_{1:T} = \{f(z_t) + \varepsilon_t\}_{t=1}^{T}$ and the latent function $f$, given by*

$$\mathcal{I}_T = \frac{1}{2} \log \det \left( \mathbf{I}_{Td} + \lambda^{-2} \mathbf{K}_T \right).$$

A detailed proof is provided in Appendix C.2. It extends the result of Srinivas et al. (2010) to the multi-output setting relevant to our analysis.

## 5 Experiments

To empirically validate the regret scaling in Theorem 2, we present synthetic experiments based on the episodic MDP setup from Section 2 to assess the performance of RL-GPS in controlled environments that reflect the assumptions in our theoretical analysis.

**Kernel complexity.** This experiment studies how kernel complexity impacts regret in synthetic MDPs generated from GP-sampled environments. The state and action spaces are continuous, $\mathcal{S} = [0, 1]^2$ and $\mathcal{A} = [0, 1]$, but each dimension is discretized into 25 equally sized bins to enable tractable value function approximation. We compare three common kernel functions: the Radial Basis Function (RBF) kernel and Matérn kernels with smoothness parameters $\nu = 2.5$ and $\nu = 1.5$. For the specific multi-output GP model, we use the popular *linear model of coregionalization* (LMC) (Grzebyk and Wackernagel, 1994; Wackernagel, 2003), which predicts the final vector-valued output as a linear combination of independent latent GPs (see Appendix D for more details).

For each kernel, a sparse LMC multi-output GP with zero mean and fixed linear correlations is used to sample the ground-truth reward and transition functions, $f_R$ and $f_S$, respectively (with rewards normalized to $r \in [0, 1]$ per step), thereby defining the MDP. The optimal value function $V^\star$ is computed using finite-horizon value iteration with $H = 20$. Algorithm 1 is run for $K = 1000$ episodes and cumulative regret relative to the optimal value function is quantified. The results are averaged over 200 randomly sampled environments and shown in Figure 1. Across all kernels, cumulative regret grows sublinearly, which is consistent with our theoretical analysis. Performance varies with the complexity of the kernel: the RBF kernel yields the lowest regret, followed by Matérn $\nu = 2.5$ and then Matérn $\nu = 1.5$, as predicted by Remark 3. This reflects that rougher kernels correspond to more complex function classes and require more data

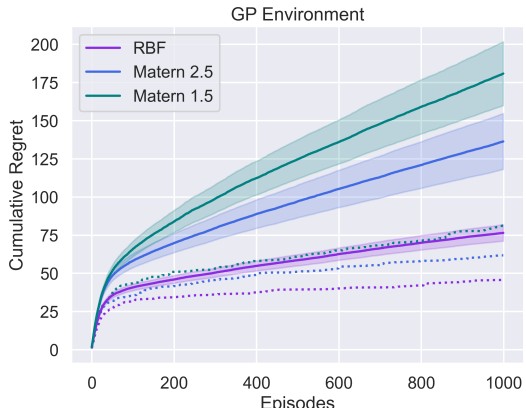

Figure 1: Cumulative regret over different kernels on GP-sampled environments over 200 trials. The shaded region around each curve represents $\pm 1$ standard error of the mean across trials. Dotted lines represent median regrets.

to accurately estimate value functions. These results empirically confirm the theoretical link between kernel smoothness and learning efficiency that our analysis formally elucidates through the regret bound's dependence on information gain $\Gamma(\cdot)$.

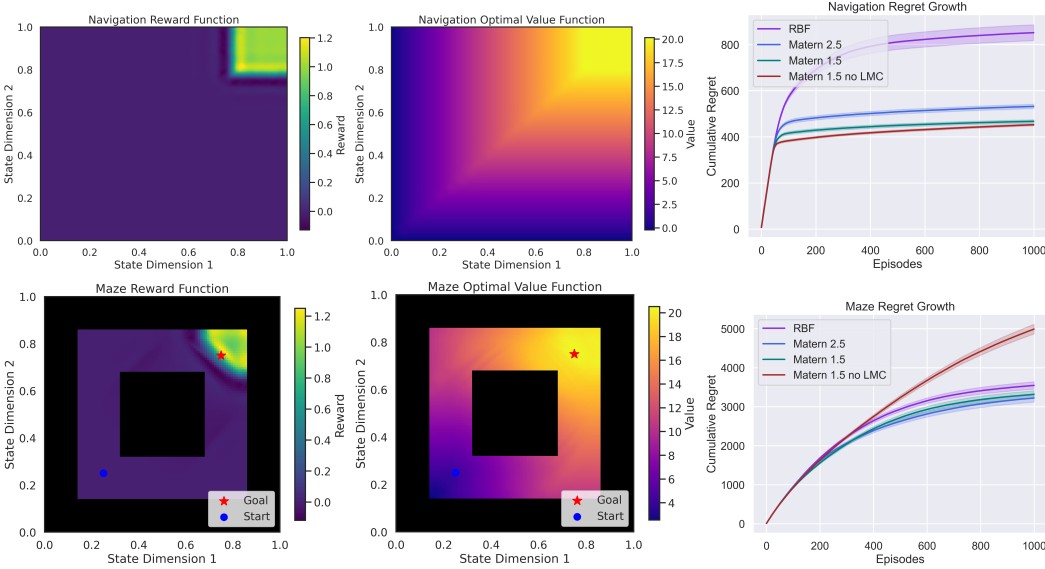

Figure 2: The reward function (left column), optimal value function (middle column), and cumulative regret of TS with a multi-output GP with confidence regions representing the standard error over 200 random trials (right column) are given. The first row corresponds to the sparse navigation task and the second row corresponds to the sparse maze problem.

**Multi-output kernel structure.** These experiments study how the multi-output kernel structure affects regret in sparse navigation tasks. The state space $\mathcal{S} = [0, 1]^2$ is discretized into 25 equally spaced bins per dimension and the action space $\mathcal{A}$ consists of 9 discrete actions that move the agent in the cardinal or diagonal directions or allow it to remain stationary. The agent receives a reward of $+1$ when within $0.1$ of the destination and a penalty of $-0.01$ otherwise. We study two settings: free movement within the grid and constrained navigation through a maze. As before, the optimal value function $V^\star$ is computed using finite-horizon value iteration. Algorithm 1 runs for $K = 1000$ episodes with a sparse LMC multi-output GP posterior. The results over 200 trials are shown in Figure 2 and demonstrate sublinear regret growth, consistent with our theory. Notably, in these sparse,

less smooth environments, the RBF kernel accumulates regret more rapidly than the Matérn kernels. This is due to the RBF kernel's strong smoothness assumptions, which lead to model misspecification and slower adaptation when the ground truth is rougher (Rasmussen and Williams, 2006). The results also highlight the value of modeling output correlations using the LMC. Specifically, the Matérn 1.5 kernel without LMC incurs substantially higher regret in the maze environment compared to its LMC-enabled counterpart. This indicates that explicitly capturing output dependencies can improve sample efficiency and reduce regret, especially in structured or high-dimensional tasks. Additional information about our experiments and results are given in Appendix E and our open-sourced implementation at: https://github.com/jbayrooti/TS_regret_study.

## 6  Conclusion

We presented a regret analysis of Thompson sampling for reinforcement learning in finite-horizon Markov Decision Processes with multi-output Gaussian Processes jointly modeling rewards and transitions. Our analysis established a sublinear regret bound of $\tilde{\mathcal{O}}(\sqrt{KH\Gamma(KH)})$, demonstrating how the complexity of the GP kernel governs learning efficiency. To derive this result, we introduced new tools for bounding uncertainty in recursive value functions, including confidence intervals for compositional functions of GPs and a multi-output elliptical potential lemma that captures correlations across components. These results extend classical GP analysis to vector-valued and recursive settings and may be of independent interest. Our experiments validated our theoretical predictions, showing sublinear regret across various tasks and kernels. Overall, this work illustrated how structural assumptions and posterior uncertainty influence the dynamics of exploration in reinforcement learning.

**Limitations.**  Our analysis relies on assumptions that may not hold in general. Specifically, we assume that the environment dynamics and rewards at each decision step are jointly Gaussian (Assumption 1) and that value functions are twice differentiable with bounded gradients and Hessians (Assumption 2). These conditions enable tractable analysis but exclude settings with discontinuous or non-smooth dynamics. Nonetheless, our smoothness assumptions are milder than those commonly imposed in prior work on regret analyses. Finally, this work focuses on finite-horizon episodic MDPs; extending the analysis to infinite-horizon settings remains a promising direction for future research.

### Acknowledgments

J. Bayrooti is supported by a DeepMind scholarship. A. Prorok is supported in part by European Research Council (ERC) Project 949940 (gAIa).

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

# A  Proof of Theorem 1

In this section, we provide a detailed proof of Theorem 1. Recall from (9) that for a single-output GP $f$ with posterior mean and standard deviation $\mu_n$ and $\sigma_n$, we have, with probability $1 - \delta$, uniformly in $z$,

$$|f(z) - \mu_n(z)| \leq \beta_n(\delta)\sigma_n(z) \tag{12}$$

where $\beta_n(\delta) = \mathcal{O}(\sqrt{\log(\frac{n}{\delta})})$.

To extend this to a composition $v(f(z))$, where $f(z) \in \mathbb{R}^d$ is drawn from a multi-output GP and $v : \mathbb{R}^d \to \mathbb{R}$ is a smooth function, we apply Taylor's theorem. For any $z \in \mathcal{Z}$, there exists a point $\zeta$ on the line segment connecting $f(z)$ and $\mu_n(z)$ such that:

$$v(f(z)) = v(\mu_n(z)) + \nabla v(\mu_n(z))^\top (f(z) - \mu_n(z)) + \frac{1}{2}(f(z) - \mu_n(z))^\top \nabla^2 v(\zeta)(f(z) - \mu_n(z)).$$

Taking absolute values and using the bounds on gradient and Hessian

$$|v(f(z)) - v(\mu_n(z))| \leq \|\nabla^\top v(\mu_n(z))\|\|f(z) - \mu_n(z)\| + \frac{1}{2}\|\nabla^2 v(\zeta)\|_{\text{op}}\|f(z) - \mu_n(z)\|^2$$

$$\leq u_G\|f(z) - \mu_n(z)\| + \frac{1}{2}u_H^2\|f(z) - \mu_n(z)\|^2$$

By the standard single GP confidence bound (9), with probability $1 - \delta/d_S$, we have $|f_j(z) - \mu_{n,j}(z)| \leq \beta_n(\delta/d_S)\sigma_{n,j}(z)$ for all $j$, and hence, applying a probability union bound, with probability $1 - \delta$,

$$\|f(z) - \mu_n(z)\| \leq \beta_n(\delta/d_s)\|\sigma_n(z)\|.$$

Substituting into the bound above, we obtain

$$|v(f(z)) - v(\mu_n(z))| \leq u_G\beta_n(\delta/d_s)\|\sigma_n(z)\| + \frac{1}{2}u_H^2\beta_n(\delta/d_s)^2\|\sigma_n(z)\|^2,$$

that completes the proof of Theorem 1.

# B  Proof of Theorem 2

In this section, we provide a detailed proof of Theorem 2 on the regret performance of RL-GPS. We bound the total regret by analyzing the per-episode difference between the optimal value function and the value function of the GP-TS-RL algorithm. We structure the proof in four main steps.

**First,** we decompose the per-step regret into two components: an immediate regret term arising from TS, and a recursive term capturing uncertainty in value propagation through the transition model.

**Second,** we bound the immediate regret using techniques inspired by those used in the analysis of TS in GP bandits. There are however certain challenges which are discussed below.

**Third,** we unroll the recursion and accumulate these bounds over all steps within an episode.

**Fourth,** we bound the cumulative regret by bounding the sum of posterior standard deviations in GPs, which appear in the third step, to complete the regret bound.

**First step: Decomposing the per-step regret.**

We begin by analyzing the regret incurred at step $h$ of episode $k$. Let $a_{h,k} = \pi_k(s_{h,k})$ denote the action taken by the algorithm, and let $a_{h,k}^\star = \arg\max_{a \in \mathcal{A}} Q_h^\star(s_{h,k}, a)$ denote the optimal action at that state. The per-step regret is defined as the difference between the optimal and executed value functions:

$$V_h^\star(s_{h,k}) - V_h^{\pi_k}(s_{h,k}).$$

By the Bellman equation, this can be written as:

$$V_h^\star(s_{h,k}) - V_h^{\pi_k}(s_{h,k})$$
$$= \left(f_R(s_{h,k}, a_{h,k}^\star) + V_{h+1}^\star(f_S(s_{h,k}, a_{h,k}^\star))\right) - \left(f_R(s_{h,k}, a_{h,k}) + V_{h+1}^{\pi_k}(f_S(s_{h,k}, a_{h,k}))\right)$$
$$= \left(f_R(s_{h,k}, a_{h,k}^\star) + V_{h+1}^\star(f_S(s_{h,k}, a_{h,k}^\star))\right) - \left(f_R(s_{h,k}, a_{h,k}) + V_{h+1}^\star(f_S(s_{h,k}, a_{h,k}))\right)$$
$$+ \left(V_{h+1}^\star(f_S(s_{h,k}, a_{h,k})) - V_{h+1}^{\pi_k}(f_S(s_{h,k}, a_{h,k}))\right)$$
$$= Q_h^\star(s_{h,k}, a_{h,k}^\star) - Q_h^\star(s_{h,k}, a_{h,k}) + \left(V_{h+1}^\star(s_{h+1,k}) - V_{h+1}^{\pi_k}(s_{h+1,k})\right), \tag{13}$$

where the first equality follows from the definition of the value function, the second adds and subtracts the term $V_{h+1}^\star(f_S(s_{h,k}, a_{h,k}))$, and the third rewrites the expression using the definition of $Q_h^\star$ and noting $s_{h+1,k} = f_S(s_{h,k}, a_{h,k})$.

We split this expression into two terms:

$$V_h^\star(s_{h,k}) - V_h^{\pi_k}(s_{h,k}) = \underbrace{Q_h^\star(s_{h,k}, a_{h,k}^\star) - Q_h^\star(s_{h,k}, a_{h,k})}_{\text{(immediate regret)}} + \underbrace{\left(V_{h+1}^\star(s_{h+1,k}) - V_{h+1}^{\pi_k}(s_{h+1,k})\right)}_{\text{(recursive part of regret)}}.$$
$$\tag{14}$$

The first term captures the immediate regret incurred by TS. The second term reflects the recursive component of regret, which arises due to uncertainty in the transition model and its impact on value propagation. We next proceed to bound the immediate regret term.

**Second step: Bounding the immediate regret (TS).**

This term captures the suboptimality of the action $a_{h,k}$ chosen by TS, relative to the optimal action $a_{h,k}^\star$, under the target function $Q_h^\star(\cdot, \cdot)$. The analysis presents two key challenges compared to the standard Thompson sampling analysis in kernel bandits (Chowdhury and Gopalan, 2017):

1. The target function $Q_h^\star(\cdot, \cdot) = f_R(\cdot, \cdot) + V_{h+1}^\star(f_S(\cdot, \cdot))$ is more complex, as it has a recursive and compositional structure involving both the reward and value functions and cannot be directly modeled as a GP.

2. The TS algorithm does not sample directly from the posterior of this target function, but instead from the posterior of a proxy $Q_h$ defined in the algorithm.

To address both challenges, we create upper and lower confidence bounds $Q^u$ and $Q^l$ for both $Q_h$ and $Q_h^\star$.

Recall the following upper and lower confidence bounds from Definition 1. We define the upper confidence bounds recursively as:

$$Q_{h,k}^u(s, a) = \mu_{R,k}(s, a) + V_{h+1,k}^u(\mu_{S,k}(s, a)) + \xi_k(s, a), \quad V_{h,k}^u(s) = \max_{a \in \mathcal{A}} Q_{h,k}^u(s, a),$$

initialized with $V_{H+1,k}^u(s) = 0$. Similarly, the lower confidence bounds are defined as:

$$Q_{h,k}^l(s, a) = \mu_{R,k}(s, a) + V_{h+1,k}^l(\mu_{S,k}(s, a)) - \xi_k(s, a), \quad V_{h,k}^l(s) = \max_{a \in \mathcal{A}} Q_{h,k}^l(s, a).$$

initialized with $V_{H+1,k}^l(s) = 0$. The confidence width $\xi_k(s, a)$ is given by:

$$\xi_k(s, a) = \beta_k(\delta/Td)\sigma_{R,k}(s, a) + G\beta_k(\delta/Td)\|\sigma_{S,k}(s, a)\| + \frac{1}{2}H\beta_k(\delta/Td)^2\|\sigma_{S,k}(s, a)\|^2, \tag{15}$$

which is based on the confidence bound from Theorem 1.

Also recall Corollary 1. With probability at least $1 - \delta$, the optimal and proxy value functions are bounded by the high-probability confidence intervals:

$$Q_{h,k}^l(s, a) \le Q_{h,k}(s, a), \quad Q_h^\star(s, a) \le Q_{h,k}^u(s, a),$$

for all $(h, k, s, a)$. Let us denote this event as $\mathcal{E}$.

Following the analysis technique used in the kernel bandit setting (Chowdhury and Gopalan, 2017), we aim to show:

$$Q_h^\star(s_{h,k}, a_{h,k}^\star) - Q_h^\star(s_{h,k}, a_{h,k}) \leq c\,\xi_{k-1}(s_{h,k}, a_{h,k}), \tag{16}$$

for some universal constant $c > 0$.

To this end, we define the *saturated set* of actions at step $(h, k)$ as:

$$\mathcal{S}_{h,k} := \left\{ a \in \mathcal{A} \ : \ Q_h^\star(s_{h,k}, a_{h,k}^\star) - Q_h^\star(s_{h,k}, a) > 2\,\xi_{k-1}(s_{h,k}, a) \right\}. \tag{17}$$

Intuitively, $\mathcal{S}_{h,k}$ includes actions that are significantly suboptimal under the true value function $Q_h^\star$, with a suboptimality gap that exceeds twice their confidence width.

We now prove a loose lower bound on the probability of selecting an action from an unsaturated set. Specifically:

$$
\begin{aligned}
\Pr\left[a_{h,k} \notin \mathcal{S}_{h,k}\right] &\geq \Pr\left[Q_{h,k}(s_{h,k}, a_{h,k}^\star) > Q_{h,k}(s_{h,k}, a) \quad \forall a \in \mathcal{S}_{h,k}\right] \\
&\geq \Pr\left[Q_{h,k}(s_{h,k}, a_{h,k}^\star) > Q_h^\star(s_{h,k}, a_{h,k}^\star) \wedge \right. \\
&\qquad \left. Q_h^\star(s_{h,k}, a_{h,k}^\star) > Q_{h,k}(s_{h,k}, a), \ \forall a \in \mathcal{S}_{h,k}\right] \\
&\geq \Pr\left[Q_{h,k}(s_{h,k}, a_{h,k}^\star) > Q_h^\star(s_{h,k}, a_{h,k}^\star)\right] - \Pr\left[\bar{\mathcal{E}}\right] \\
&\geq \frac{1}{2} - \delta.
\end{aligned}
$$

The first inequality holds because $a_{h,k}^\star$ is, by definition, the optimal action under $Q_h^\star$ and therefore saturated. The second step follows by decomposing the event into two sufficient conditions. To see the third line, observe that under the event $\mathcal{E}$, for any saturated action $a \in \mathcal{S}_{h,k}$, we have:

$$Q_{h,k}(s_{h,k}, a) \leq Q_h^\star(s_{h,k}, a) + 2\xi_{k-1}(s_{h,k}, a) \leq Q_h^\star(s_{h,k}, a_{h,k}^\star).$$

The fourth inequality follows from the symmetry and probability bound for $\mathcal{E}$ from Corollary 1.

Let $b_{h,k} = \arg\min_{a \in \mathcal{A} \setminus \mathcal{S}_{h,k}} \xi_{k-1}(s_{h,k}, a)$ denote the unsaturated action with the smallest confidence width. Then, using the law of total expectation:

$$
\begin{aligned}
\mathbb{E}\left[\xi_{k-1}(s_{h,k}, a_{h,k})\right] &\geq \mathbb{E}\left[\xi_{k-1}(s_{h,k}, a_{h,k}) \mid a_{h,k} \notin \mathcal{S}_{h,k}\right] \Pr\left[a_{h,k} \notin \mathcal{S}_{h,k}\right] \\
&\geq \xi_{k-1}(s_{h,k}, b_{h,k})(\frac{1}{2} - \delta). \tag{18}
\end{aligned}
$$

We now upper bound the immediate regret using $b_{h,k}$ as a reference:

$$
\begin{aligned}
&Q_h^\star(s_{h,k}, a_{h,k}^\star) - Q_h^\star(s_{h,k}, a_{h,k}) \\
&= \left(Q_h^\star(s_{h,k}, a_{h,k}^\star) - Q_h^\star(s_{h,k}, b_{h,k})\right) + \left(Q_h^\star(s_{h,k}, b_{h,k}) - Q_h^\star(s_{h,k}, a_{h,k})\right) \\
&\leq 2\,\xi_{k-1}(s_{h,k}, b_{h,k}) + \left(Q_{h,k}(s_{h,k}, b_{h,k}) + \xi_{k-1}(s_{h,k}, b_{h,k})\right) \\
&\quad - \left(Q_{h,k}(s_{h,k}, a_{h,k}) - \xi_{k-1}(s_{h,k}, a_{h,k})\right) \\
&\leq 3\,\xi_{k-1}(s_{h,k}, b_{h,k}) + \xi_{k-1}(s_{h,k}, a_{h,k}) \\
&\leq \left(\frac{3}{\frac{1}{2} - \delta} + 1\right) \xi_{k-1}(s_{h,k}, a_{h,k}),
\end{aligned}
$$

where the first inequality adds and subtract the value at $b_{h,k}$, the first inequality uses definitions of the set $\mathcal{S}$, $b_{h,k}$ and event $\mathcal{E}$ and the final step uses the earlier bound on $\xi_{k-1}(s_{h,k}, b_{h,k})$.

This completes the bound on immediate regret in terms of the confidence width at the selected action.

**Third step: Bounding the episode regret.**

From the per-step regret decomposition (14) and the bound on the immediate regret (16) and using a telescoping sum over steps $h = 1$ to $H$, we obtain the following bound on the regret incurred in episode $k$:

$$V_1^\star(s_{1,k}) - V_1^{\pi_k}(s_{1,k}) \leq c \sum_{h=1}^{H} \xi_{k-1}(s_{h,k}, a_{h,k}).$$

**Fourth step: Bounding the total regret.**

Summing the episode regret over $k = 1$ to $K$ episodes, we have:

$$\text{Regret}(T) \leq c \sum_{k=1}^{K} \sum_{h=1}^{H} \xi_{k-1}(s_{h,k}, a_{h,k}).$$

Replacing $\xi_k$, we get:

$$\text{Regret}(T) \leq \beta_K(\delta/Td) \sum_{k=1}^{K} \sum_{h=1}^{H} \left( \sigma_{R,k-1}(s_{h,k}, a_{h,k}) + u_G \left\| \sigma_{S,k-1}(s_{h,k}, a_{h,k}) \right\| \right)$$

$$+ \frac{u_H \beta_K^2(\delta/Td)}{2} \sum_{k=1}^{K} \sum_{h=1}^{H} \left\| \sigma_{S,k-1}(s_{h,k}, a_{h,k}) \right\|^2 \tag{19}$$

The sum of sequentially conditioned standard deviations of a sequence of observations from a GP often appears in the analysis of regret bounds in Bayesian optimization. A classical result by Srinivas et al. (2010) shows that in the case of a single output GP, for a sequence of inputs $z_1, z_2, \ldots, z_T$, we have:

$$\sum_{i=1}^{n} \sigma_{i-1}^2(z_i) \leq C \log \det \left( \mathbf{I}_n + \frac{1}{\lambda^2} \mathbf{K}_n \right),$$

where $\lambda$ is the GP noise parameter, $\mathbf{K}_n$ is the kernel matrix over the observed inputs and $C = 2/\log(1 + 1/\lambda^2)$ is a constant. This result is sometimes referred to as the elliptical potential lemma, especially in the special case of linear kernels (Carpentier et al., 2020).

A direct application of this result to our setting faces two key challenges: *i)* We model transitions using a multi-output GP. Naively applying the bound to each output dimension separately results in a regret bound that scales suboptimally with $d_S$, the dimension of the state space. *ii)* Our regret decomposition involves a double sum over episodes $k$ and steps $h$, but within each episode, the observations across $h$ are not sequential updates. This structure leads to an additional scaling with the episode length $H$.

We address both challenges. First, we derive a new elliptical potential lemma tailored for multi-output GPs, which improves the dependence on $d_S$ in the regret bound. That is given in Lemma 1. Second, to improve the $H$ dependence, we leverage tools and techniques from the analysis of GPs under batch observations and delayed feedback (Calandriello et al., 2020) to tighten the bound with respect to $H$ (that roughly speaking can be understood as delay in updating the GP model).

**Lemma 2** (Elliptical potential for multi-output GPs with delayed updates). *Let* $f : \mathcal{Z} \to \mathbb{R}^d$ *be a $d$-dimensional function modeled as a multi-output GP with a matrix-valued kernel $k$. Let* $z_1, \ldots, z_T \in \mathcal{Z}$ *be a sequence of input points and suppose the GP posterior is only updated every $H$ steps, so that the standard deviation at time $t$ is $\sigma_{H\lfloor(t-1)/H\rfloor}(z_t) \in \mathbb{R}^d$. Then,*

$$\sum_{t=1}^{T} \left\| \sigma_{H\lfloor(t-1)/H\rfloor}(z_t) \right\| \leq \sqrt{\frac{4\Gamma(T)}{\log(1 + \lambda^{-2})} \left( T + \frac{4H^2 \Gamma(T/H)}{\log(1 + \lambda^{-2})} \right)}.$$

Applying Lemma 2 to the regret bound in terms of uncertainties (19), we obtain

$$\text{Regret}(T) = \mathcal{O} \left( \log(Td/\delta) \sqrt{T\Gamma(T)} \right). \tag{20}$$

## C  Auxiliary proofs

### C.1  Proof of Corollary 1

We prove the lower bound by induction; the upper bound can be shown similarly.

As the base case, observe that $V_{H+1} = V_{H+1}^\star = V_{H+1}^l = 0$.

Now consider the inductive step. We compare the value of $Q_{h,k}^l(s,a)$ to $Q_h^\star(s,a)$:

$$Q_{h,k}^l(s,a) - Q_h^\star(s,a) = \mu_{R,k}(s,a) + V_{h+1,k}^l(\mu_{S,k}(s,a)) - \xi_k(s,a) - \left(f_R(s,a) + V_{h+1}^\star(f_S(s,a))\right)$$

$$= \underbrace{\mu_{R,k}(s,a) - f_R(s,a) + V_{h+1}^\star(\mu_{S,k}(s,a)) - V_{h+1}^\star(f_S(s,a)) - \xi_k(s,a)}_{\text{Term 1}}$$

$$+ \underbrace{V_{h+1,k}^l(\mu_{S,k}(s,a)) - V_{h+1}^\star(\mu_{S,k}(s,a))}_{\text{Term 2}}$$

$$\leq 0.$$

The first line follows from the definitions of $Q_{h,k}^l$ and $Q_h^\star$. The second line is obtained by adding and subtracting $V_{h+1}^\star(\mu_{S,k}(s,a))$, followed by regrouping terms. The inequality holds since Term 1 is non-positive due to the confidence bounds in Theorem 1, and Term 2 is non-positive by the induction hypothesis.

Next, we extend the argument to the value function:

$$V_h^l(s) - V_h^\star(s) = \max_{a \in \mathcal{A}} Q_h^l(s,a) - \max_{a \in \mathcal{A}} Q_h^\star(s,a)$$

$$\leq \max_{a \in \mathcal{A}} \left[Q_h^l(s,a) - Q_h^\star(s,a)\right]$$

$$\leq 0,$$

which completes the inductive proof that $Q_{h,k}^l(s,a) \leq Q_h^\star(s,a)$ and $V_h^l(s) \leq V_h^\star(s)$ for all $s,a,h$.

We know that $Q_{h,k}^l(s,a) \leq Q_{h,k}(s,a)$ for all $(s,a,h,k)$. For the inductive step, consider,

$$Q_{h,k}^l(s,a) - Q_{h,k}(s,a)$$

$$= \mu_{R,k}(s,a) + V_{h+1,k}^l(\mu_{S,k}(s,a)) - \xi_k(s,a) - \left(f_{R,k}(s,a) + V_{h+1,k}(f_{S,k}(s,a))\right)$$

$$= \underbrace{\mu_{R,k}(s,a) - f_{R,k}(s,a) + V_{h+1,k}(\mu_{S,k}(s,a)) - V_{h+1,k}(f_{S,k}(s,a)) - \xi_k(s,a)}_{\text{Term 1}}$$

$$+ \underbrace{V_{h+1,k}^l(\mu_{S,k}(s,a)) - V_{h+1,k}(\mu_{S,k}(s,a))}_{\text{Term 2}}$$

$$\leq 0.$$

The decomposition follows by adding and subtracting $V_{h+1,k}(\mu_{S,k}(s,a))$ and regrouping terms. Term 1 is non-positive due to the high-probability confidence bound (Theorem 1). Term 2 is non-positive by the induction hypothesis.

Hence, we conclude $Q_{h,k}^l(s,a) \leq Q_{h,k}(s,a)$ for all $(s,a)$.

Extending to the value function:

$$V_{h,k}^l(s) - V_{h,k}(s) = \max_{a \in \mathcal{A}} Q_{h,k}^l(s,a) - \max_{a \in \mathcal{A}} Q_{h,k}(s,a)$$

$$\leq \max_{a \in \mathcal{A}} \left(Q_{h,k}^l(s,a) - Q_{h,k}(s,a)\right)$$

$$\leq 0.$$

This completes the inductive proof that $Q_{h,k}^l(s,a) \leq Q_{h,k}(s,a)$ and $V_{h,k}^l(s) \leq V_{h,k}(s)$ for all $s,a,h$. The upper bounds, i.e., $Q_h^\star(s,a), Q_{h,k}(s,a) \leq Q_{h,k}^u(s,a)$ for all $s,a,h$, are proven analogously using similar argument.

## C.2 Proof of Lemma 1

We consider a $d$-dimensional GP $f: \mathcal{Z} \to \mathbb{R}^d$ and a sequence of inputs $z_1, \ldots, z_T$. Define the full observation vector as:

$$y_{1:T} = \begin{bmatrix} f(z_1) + \varepsilon_1 \\ \vdots \\ f(z_T) + \varepsilon_T \end{bmatrix} \in \mathbb{R}^{Td}, \quad \varepsilon_t \sim \mathcal{N}(0, \lambda^2 I_d).$$

The mutual information between $y_{1:T}$ and the latent function values is:

$$\mathcal{I}_T = I(y_{1:T}; f(z_1), \ldots, f(z_T)) = \frac{1}{2} \log \det \left( \mathbf{I}_{Td} + \lambda^{-2} \mathbf{K}_T \right),$$

where $\mathbf{K}_T \in \mathbb{R}^{Td \times Td}$ is the prior kernel matrix over all outputs.

Let $\sigma_{t-1}(z_t) \in \mathbb{R}^d$ be the vector of posterior standard deviations at $z_t$ given observations up to time $t-1$. Define the total uncertainty:

$$S_T := \sum_{t=1}^{T} \|\sigma_{t-1}(z_t)\|^2 = \sum_{t=1}^{T} \sum_{j=1}^{d} \sigma_{t-1,j}^2(z_t).$$

We apply the scalar inequality (used in Srinivas et al. (2010)):

$$\sigma^2 \leq \frac{1}{\log(1 + \lambda^{-2})} \log\left(1 + \frac{\sigma^2}{\lambda^2}\right), \quad \text{for all } \sigma^2 \in [0, 1].$$

Applying this to each term $\sigma_{t-1,j}^2(z_t)$ and summing, we obtain:

$$S_T \leq \frac{1}{\log(1 + \lambda^{-2})} \sum_{t=1}^{T} \sum_{j=1}^{d} \log\left(1 + \frac{\sigma_{t-1,j}^2(z_t)}{\lambda^2}\right).$$

Since $y_{1:T}$ is jointly Gaussian, the total sum of these log-terms is bounded by $2\mathcal{I}_T$, giving:

$$S_T \leq \frac{2}{\log(1 + \lambda^{-2})} \mathcal{I}_T.$$

Thus, we conclude:

$$\sum_{t=1}^{T} \|\sigma_{t-1}(z_t)\|^2 \leq C \mathcal{I}_T, \quad \text{with } C = \frac{2}{\log(1 + \lambda^{-2})}.$$

### C.3 Proof of Lemma 2

We begin by naively applying the elliptical potential lemma to the same step index across episodes

$$\sum_{t=1}^{T} \|\sigma_{H\lfloor (t-1)/H \rfloor}(z_t)\| \leq \sum_{h=1}^{H} \sum_{j=1}^{K} \|\sigma_{H(j-1)+h}(z_{Hj+h})\|^2$$

$$\leq CH\Gamma(K) \tag{21}$$

To improve on this, we use the following inequality proven in Lemma 3, for any $z$ and $t' < t$,

$$\|\sigma_{t'}(z)\|^2 \leq \|\sigma_t(z)\|^2 \left(1 + \sum_{j=t'+1}^{t} \|\sigma_{t'}(z_j)\|^2\right).$$

Applying this to the case where the model is updated every $H$ steps, for each $t \in [T]$, we let $t' = H\lfloor (t-1)/H \rfloor$. Then,

$$\|\sigma_{t'}(z_t)\| \leq \|\sigma_t(z_t)\| \sqrt{1 + \sum_{j=t'+1}^{t} \|\sigma_{t'}(z_j)\|^2}.$$

Now, summing over all $t = 1, \ldots, T$, we apply the Cauchy–Schwarz inequality:

$$\sum_{t=1}^{T} \|\sigma_{H\lfloor (t-1)/H \rfloor}(z_t)\| \leq \sum_{t=1}^{T} \|\sigma_t(z_t)\| \sqrt{1 + \sum_{j=H\lfloor (t-1)/H \rfloor+1}^{t} \|\sigma_{H\lfloor (t-1)/H \rfloor}(z_j)\|^2}$$

$$\leq \left(\sum_{t=1}^{T} \|\sigma_t(z_t)\|^2\right)^{1/2} \left(T + H \sum_{t=1}^{T} \|\sigma_{H\lfloor (t-1)/H \rfloor}(z_t)\|^2\right)^{1/2}.$$

We now substitute the same term with the looser bound given earlier in (21),

$$\sum_{t=1}^{T} \|\sigma_{H\lfloor(t-1)/H\rfloor}(z_t)\| \leq \sqrt{\frac{2\Gamma(T)}{\log(1+\lambda^{-2})}\left(T + \frac{2H^2\Gamma(K)}{\log(1+\lambda^{-2})}\right)}.$$

This completes the proof.

**Lemma 3** (Variance ratio inequality for multi-output GPs). *Let $f : \mathcal{Z} \to \mathbb{R}^d$ be a d-dimensional function modeled as a multi-output GP with a matrix-valued kernel $k$. Let $\sigma_t(z) \in \mathbb{R}^d$ denote the vector of posterior marginal standard deviations at point $z$ given $t$ observations. Then, for any $z \in \mathcal{Z}$ and $t' < t$,*

$$1 \leq \frac{\|\sigma_{t'}(z)\|^2}{\|\sigma_t(z)\|^2} \leq 1 + \sum_{j=t'+1}^{t} \|\sigma_{t'}(z_j)\|^2.$$

*Proof.* For each output dimension $j \in [d]$, the scalar variance update satisfies (see, e.g., Lemma 4 of Calandriello et al. (2020) and Proposition A.1 of Calandriello et al. (2022))

$$\frac{\sigma_{t',j}^2(z)}{\sigma_{t,j}^2(z)} \leq 1 + \sum_{i=t'+1}^{t} \sigma_{t',j}^2(z_i).$$

Now summing over $j = 1, \ldots, d$ we get:

$$\frac{\|\sigma_{t'}(z)\|^2}{\|\sigma_t(z)\|^2} = \frac{\sum_{j=1}^{d} \sigma_{t',j}^2(z)}{\sum_{j=1}^{d} \sigma_{t,j}^2(z)} \leq 1 + \sum_{i=1}^{d}\sum_{j=t'+1}^{t} \sigma_{t',i}^2(z_j) = 1 + \sum_{j=t'+1}^{t} \|\sigma_{t'}(z_j)\|^2.$$

Therefore,

$$\frac{\|\sigma_{t'}(z)\|^2}{\|\sigma_t(z)\|^2} \leq 1 + \sum_{j=t'+1}^{t} \|\sigma_{t'}(z_j)\|^2.$$

The lower bound $1 \leq \|\sigma_{t'}(z)\|^2/\|\sigma_t(z)\|^2$ holds since variance decreases monotonically as more data is observed. This completes the proof. $\square$

## D  Discussion on the linear model of coregionalization

A widely used and computationally convenient special case of multi-output GPs is the *linear model of coregionalization* (LMC) (Grzebyk and Wackernagel, 1994; Wackernagel, 2003). In this model, the vector-valued function $f : \mathcal{Z} \to \mathbb{R}^d$ is expressed as a linear combination of $L$ independent latent Gaussian processes:

$$f_j(z) = \sum_{\ell=1}^{L} \alpha_{j\ell} g_\ell(z), \quad g_\ell \sim \text{GP}(0, k^{(g)}), \tag{22}$$

where $k^{(g)} : \mathcal{Z} \times \mathcal{Z} \to \mathbb{R}$ is a shared scalar kernel, and $\alpha \in \mathbb{R}^{d \times L}$ is a matrix of output mixing weights. This induces a matrix-valued kernel:

$$k(z, z') = \mathbf{A}\,k^{(g)}(z, z'), \quad \text{where } \mathbf{A} = \alpha\alpha^\top \in \mathbb{R}^{d \times d}. \tag{23}$$

Under this kernel structure, the block kernel matrix over the training data admits a Kronecker product decomposition:

$$\mathbf{K}_n = \mathbf{A} \otimes \mathbf{K}_n^{(g)}, \tag{24}$$

where $\mathbf{K}_n^{(g)} \in \mathbb{R}^{n \times n}$ is the input kernel matrix with $[\mathbf{K}_n^{(g)}]_{ij} = k^{(g)}(z_i, z_j)$. The cross-covariance matrix between test point $z$ and training data becomes:

$$\mathbf{k}_n(z) = \mathbf{A} \otimes \mathbf{k}_n^{(g)}(z) \in \mathbb{R}^{nd \times d}, \tag{25}$$

with $[\mathbf{k}_n^{(g)}(z)]_i = k^{(g)}(z_i, z)$.

This formulation is particularly useful when jointly modeling structured outputs such as reward and transition functions in reinforcement learning, as it captures both intra- and inter-output correlations while enabling scalable inference. We provide a brief discussion on the regret bounds with such a structured kernel.

## D.1 Information gain and regret bounds for LMC

We analyze how the structure of the LMC affects the information gain term $\Gamma(T)$ appearing in the regret bound. Recall that $\Gamma(T)$ upper bounds the quantity $\frac{1}{2} \log \det(\mathbf{I}_{Td} + \frac{1}{\lambda^2} \mathbf{K}_T)$, where $\mathbf{K}_T$ is the kernel matrix of the multi-output GP. Under the LMC structure, $\mathbf{K}_T = \mathbf{A} \otimes \mathbf{K}_T^{(g)}$, where $\mathbf{A}$ captures output correlations and $\mathbf{K}_T^{(g)}$ is the kernel matrix corresponding to a shared latent GP. Using properties of Kronecker products and letting $\lambda_i$ denote the eigenvalues of $\mathbf{A}$, we obtain:

$$\log \det \left( \mathbf{I}_{Td} + \frac{1}{\lambda^2} \mathbf{K}_T \right) = \log \det \left( \mathbf{I}_{Td} + \frac{1}{\lambda^2} \mathbf{A} \otimes \mathbf{K}_T^{(g)} \right)$$
$$= \sum_{i=1}^{d} \log \det \left( \mathbf{I}_T + \frac{\lambda_i}{\lambda^2} \mathbf{K}_T^{(g)} \right).$$

For the Matérn family of kernels, the information gain of the latent scalar GP is known to satisfy (Vakili et al., 2021a; Whitehouse et al., 2023):

$$\Gamma^{(g)}(T) = \tilde{\mathcal{O}} \left( \left( \frac{T}{\lambda^2} \right)^{\alpha} \right),$$

where $\alpha = \frac{d}{2\nu + d} < 1$ depends on the input dimension $d$ and the kernel smoothness parameter $\nu$.

Substituting into the sum, we obtain:

$$\Gamma(T) = \sum_{i=1}^{d} \tilde{\mathcal{O}} \left( \left( \frac{T\lambda_i}{\lambda^2} \right)^{\alpha} \right)$$
$$= \tilde{\mathcal{O}} \left( \left( \sum_{i=1}^{d} \lambda_i^{\alpha} \right) \left( \frac{T}{\lambda^2} \right)^{\alpha} \right). \tag{26}$$

In the general case without structure, a standard upper bound is given by:

$$\Gamma(T) = \tilde{\mathcal{O}} \left( \left( \frac{Td}{\lambda^2} \right)^{\alpha} \right).$$

Comparing the two, we observe that when $\mathbf{A}$ has low-rank behavior, specifically, when $\sum_{i=1}^{d} \lambda_i^{\alpha} \leq d^{\alpha}$, the LMC-based bound in (26) can be tighter. In particular, the regret bound becomes:

$$\text{Regret} = \tilde{\mathcal{O}} \left( \left( \sum_{i=1}^{d} \lambda_i^{\frac{d}{2\nu + d}} \right) T^{\frac{d}{2\nu + d}} \right). \tag{27}$$

This shows the effect of shared latent structure and output correlations on the regret bounds.

# E  Additional experiments

In this section, we provide further information about our experiments and additional results. Note that the code we used to run these experiments is available at https://github.com/jbayrooti/TS_regret_study.

**Model training.**  We model the reward and transition functions using a multi-output sparse variational Gaussian process, trained by maximizing the evidence lower bound (ELBO) with the Adam optimizer (Diederik and Ba, 2015). A shared base kernel (either RBF or Matérn) is used across outputs, and the outputs are linearly mixed according to a matrix following a linear model of coregionalization (LMC). The model uses 100 inducing points per output dimension, a zero mean function, and is optimized for 20 steps per iteration using GPyTorch (Gardner et al., 2018). Full code is included in the supplementary material for reproducibility.

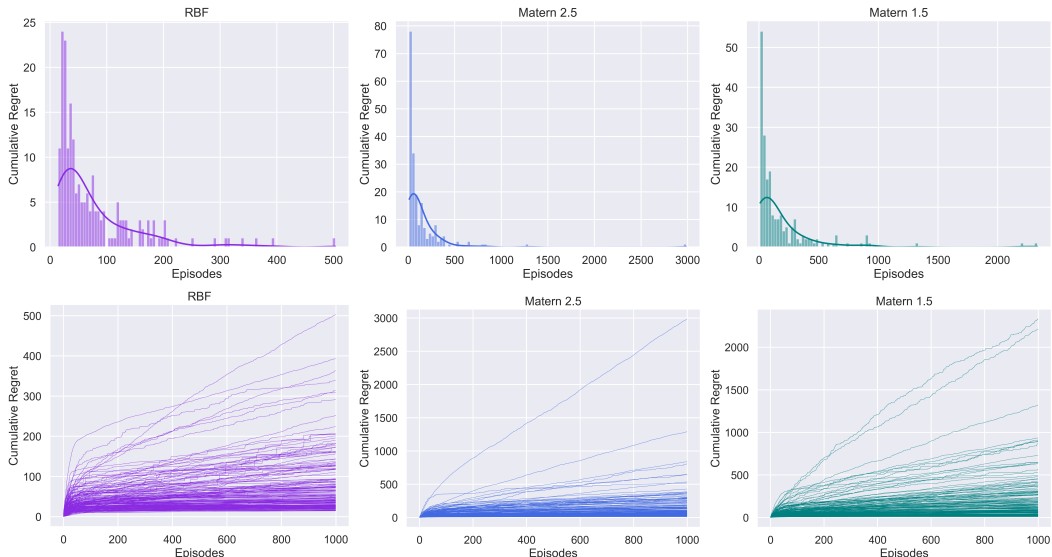

Figure 3: The first row shows the histogram of cumulative regrets over all trials on the last episode of RL-GPS training with each kernel. The second row shows the regret growth curves from all trials for each kernel.

**Kernel complexity.** We study the effect of kernel complexity on regret using synthetic MDPs generated from 200 different GP-sampled environments. For all GPs, we fix a random linear mixing matrix (see Appendix D),

$$\mathbf{A} = \begin{pmatrix} 0.9926 & 0.2082 & 0.4968 \\ -0.3196 & 0.8869 & 0.1603 \\ 0.1557 & -1.4231 & -1.3905 \end{pmatrix}.$$

In each trial, we sample new ground-truth reward and transition functions from the multi-output GP to define an MDP. The optimal value function is then computed for this MDP via finite-horizon value iteration with horizon $H = 20$. We train RL-GPS (Algorithm 1) for $K = 1000$ episodes in each environment and record the cumulative regret. Since the mixing matrix is fixed, randomness within a trial arises only from the starting state distribution during rollouts. Across trials, randomness stems from variation in the sampled environments, which can differ significantly in difficulty. As a result, we observe a substantial right-skew in cumulative regret, with a few environments producing particularly challenging instances. To demonstrate this variability, we show the cumulative regret at the final episode across all trials for each kernel as well as the individual regret curves for each trial in Figure 3. All trials for each experiment run within 24 hours on one NVIDIA GeForce RTX 2080 Ti GPU.

**Multi-output kernel structure.** For this set of experiments, we train the multi-output GP using a Matérn kernel with $\nu = 1.5$, comparing two modeling approaches: independent GPs and the linear model of coregionalization (LMC). When using LMC, the mixing weights are learned during training, so the randomness across trials arises only from the random initialization of the mixing weights and the starting states for rollouts. In contrast to the earlier experiments, the environment itself is fixed across all trials. For the unconstrained navigation experiments, the horizon used is $H = 20$ and the GP is updated for 20 optimization steps at each iteration. For the maze experiments, the horizon used is also $H = 20$ and, due to the increased modeling difficulty, the GP is updated for 50 optimization steps at each iteration. All trials for each unconstrained navigation experiment run within 24 hours on one NVIDIA GeForce RTX 2080 Ti GPU. Due to the greater number of GP updates in the maze setting, trials take longer to run and all 200 trials complete within 80 hours on the same hardware. Note that all experiments are easily parallelizable.

