# OpenReview forum: "No-Regret Thompson Sampling for Finite-Horizon Markov Decision Processes with Gaussian Processes"
_NeurIPS.cc/2025/Conference — NeurIPS 2025 poster_

### Official Review · Reviewer_oQJt · 2025-06-11

**Clarity:** 3
**Significance:** 3
**Originality:** 3
**Rating:** 5
**Confidence:** 2

**Summary:**

The paper showcases new bounds for Thompson Sampling algorithms in Reinforcement Learning under Gaussian Process assumptions for the rewards and transition functions. It considers the case where the a multitask GP is used to generate reward and transition functions, which at its simplest case covers Bayesian optimisation, as well as more complex RL tasks.

The first challenge that the paper faces is the fact that the optimal value function is not a GP but a composition of GP (deep GP). Theorem 1 provides confidence bounds for the this case, where the bounds depend on the standard deviation up to a quadratic term, with constants depending over the gradient and hessian of the value function.

Using this, Theorem 2 is proved which bounds the regret of the algorithm, achieving bounds of: $$ \mathcal{O}(\log(T)d / \delta) \sqrt{T \Gamma(T)} $$

where $d$ is the output dimensionality, $\delta$ the probability that the bound does not hold, and $\gamma$ an upper bound on the information gain.

The proof follows the steps:

1. Decomposing the regret into an immediate step, and a recursive part.

2. Bounding the immediate regret. This is done by considering the confidence bounds on the value functions which are compositional.

3. The regret then gets accumulated across each episode (i.e. for $K$ episodes of length $H$, we accumulate accumulate into $K$ parts.

4. The sum of all $K$ per-episode regrets are bounded.

The individual regrets are using the elliptical potential lemma for multi-output GPs, which extends a classical result that sums of variances of sequences of inputs can be bounded by the log-determinant term. The extension allows for tighter bounds when considering multi-output GPs.

The theoretical findings are then empirically validated.

**Questions:**

- Are there any real-life examples where the algorithm has been used in the literature?

**Ethical Concerns:**

["NO or VERY MINOR ethics concerns only"]

**Final Justification:**

Having read the other reviews and rebuttal, I maintain my original decision to recommend acceptance. This stems from my belief that theoretical contributions are difficult to obtain and should be valued, even if they will naturally cover smaller application areas.

**Limitations:**

Limitations are discussed at the end of the paper.

**Quality:**

3

**Strengths And Weaknesses:**

Strengths:

- Quality: The paper appears to be technically sound and there are no errors I could find.

- Clarity: I found the paper very easy to read and well structured, I was able to follow all arguments even if I am do not have much experience with theoretical work. Clarity on the experimental section could be improved though.

- Significance: Regret bounds are always an important development, and it is good to obtain them for Thompson Sampling, since many algorithms use it. This work can serve as a stepping stone to future bounds for new algorithms.

- Originality: The bounds appear to be novel.

Weaknesses:

The experimental section is not written clear enough, as it is difficult to understand the link between the results and the theoretical findings. In short: the figures tell very little without referring to the large amount of text, it would be nice to find a way a visual way of showing the link between the experiments and the empirical results.

---

> ### Author Rebuttal · Authors · 2025-07-31
>
> Thank you very much for reviewing our paper and for your positive feedback. We are pleased to hear that you found our regret bounds to be a valuable contribution to the community and appreciate the practical recommendations. We address your comments below and hope this will help improve your evaluation of the paper.
>
> **Weaknesses**
>
> 1. *Unclear experimental section*. \
> Thank you for highlighting this. We agree that a more detailed explanation of the link between the experiments and theoretical analysis would improve clarity. Our experiments are specifically designed to validate our theoretical findings by confirming the sublinear growth of cumulative regret, consistent with our $\mathcal{\tilde{O}}(\sqrt{KH\Gamma(KH)})$ bound, and by illustrating how the choice of GP kernel influences learning efficiency in line with the theoretical dependence on $\Gamma(\cdot)$. To emphasize these connections, we will add approximate theoretical reference curves to Figures 1 and 2 in the camera-ready version to visually highlight the sublinear regret growth predicted by our theory relative to the regret growth in practice. We are also open to incorporating additional visualizations if recommended (although we are constrained by this year’s rebuttal format and cannot upload revised figures at this stage).
>
> **Questions**
>
> 1. *Are there examples of the algorithm in literature?* \
> Yes, the algorithm we study is closely related to a recent approach called HOT-GP (Bayrooti et al., 2025), which achieved strong empirical performance on standard benchmarks such as MuJoCo and VMAS. On these tasks, HOT-GP was shown to match or outperform both model-free (e.g., PPO, SAC) and model-based (e.g., MBPO, UCB-based) methods. While this prior work showed the algorithm’s practical utility in simulated robotics settings, it did not provide theoretical guarantees. Our work abstracts the core ideas behind HOT-GP into a form that enables a rigorous regret analysis, serving as the theoretical foundation for this class of algorithms.
>
> Thanks again for your review and we would be happy to further discuss any of these points during the author-reviewer discussion period.

---

> > ### Comment · Reviewer_oQJt · 2025-08-01
> >
> > I’d like to thank the authors for their time in responding to my questions. Having read other reviews and responses, I maintain my recommendation to accept, as I see the paper as a valuable theoretical contribution to the field.

---

> > > ### Author Response · Authors · 2025-08-05
> > >
> > > Thank you for acknowledging our response and the recommendation for acceptance. We will update the presentation of the experimental section based on the discussion above, which we believe will further strengthen the paper. Thank you again for your time reviewing our work.

---

### Official Review · Reviewer_7dz9 · 2025-06-28

**Clarity:** 3
**Significance:** 2
**Originality:** 1
**Rating:** 4
**Confidence:** 4

**Summary:**

The paper analyzes Thompson Sampling (TS) in episodic finite-horizon Markov-Decision Processes whose reward and transition dynamics are modeled jointly as a multi-output Gaussian Process (GP), and proves a sublinear regret bound of $\tilde{\mathcal{O}}(\sqrt{K H \Gamma(K H)})$ over $K$ episodes of horizon $H$,
where $\Gamma(\cdot)$ is the complexity of the GP model.  This paper demonstrates the confidence bounds for composed GPs and extend traditional elliptical potential lemma to multi-output cases. The regret bound is validated by numerical experiments.

**Questions:**

1. Could you explain more on the novelty of the paper? The idea follows Posterior-Sampling RL (PSRL).  Moving from two  GPs to one multi-output GP is not very novel. Moreover, most  proofs of the main results look standard or straightforward. The proof of Theorem  1 is a direct application of Taylor expansion and confidence bounds for the standard GP. The proof of multi-output elliptical potential lemma  seems to be a simple extension of standard elliptical potential lemma. Could you explain the novelty and key part in theoretical analysis?

2. The numerical experiments are on small-scale problems with $\mathcal{S}=[0,1]^2$ and $\mathcal{A}=[0,1]$. This toy example is small. My concern lies in the scalability with respect to the cardinality and dimension of the state and action spaces. Moreover, there is no comparison with other benchmarks.
3. Could you offer a practical example where Assumptions 1-2 hold?

**Ethical Concerns:**

["NO or VERY MINOR ethics concerns only"]

**Final Justification:**

Authors clarify the technical novelty of their paper in the response, which previously was my main concern. Although their paper lacks large-scale experiments,  it may helps to future works about TS with GP.

**Limitations:**

Yes.

**Paper Formatting Concerns:**

None.

**Quality:**

3

**Strengths And Weaknesses:**

**Strengths:**

1. This paper is a complete work with both theoretical guarantees and numerical results. The regret bound is shown with rigorous proofs.
2. Paper is well structured and easy to understand.
3. Tools (Theorem 1, Lemma 1) are likely reusable in other Bayesian-RL analysis.
4. First regret bound for TS with joint GP modelling of reward and transition.

**Weakness:**
1. This paper assume strong modeling assumptions: rewards and dynamics are jointly Gaussian and value functions  are twice differentiable with bounded Hessian (Assumptions 1–2). This may be not applicable in most practical problems.
2. Experiments are on small-scale probllems (2-dimensional states, discretised) and lack comparison with other baselines.
3. There is not much novelty in the proposed idea. The proposed algorithm is a simple extension of posterior sampling RL (PSRL) by replacing posterior distribution with a Gaussian process, which is a non-parametric counterpart of posterior distribution, and most analysis procedures are standard.

---

> ### Author Rebuttal · Authors · 2025-07-31
>
> Thank you very much for your time spent reviewing our paper. We are glad that you find the work complete, the paper well-structured, and the proofs rigorous with some results reusable by the community. We address your comments in detail below and hope this will help improve your evaluation of the paper.
>
> 1. *Significance of contribution and technical novelty.* \
> Thank you for the opportunity to clarify this. At a high level, both TS and RL  are known to be inherently challenging to analyze. For instance, although TS is a natural algorithm dating back to Thompson (1933), its first rigorous analysis in the much simpler setting of K-armed bandits only appeared in the 2010s (Agrawal and Goyal, 2012; Kaufmann et al., 2012a). Extending such analyses to the RL setting, particularly with continuous state and action spaces and minimal structural assumptions, remains a significant open challenge.\
> Our work takes a step forward in this direction by providing, to the best of our knowledge, the first sublinear regret bound for TS in continuous state-action MDPs using multi-output GP modeling of the reward and transition functions. We build on tools from the literature while also introducing key theoretical contributions to overcome challenges in this setting through a modular proof strategy. We highlight these innovations by walking through our analysis:
>    - Step 1: We decompose the regret into an immediate regret term and a recursive term that captures uncertainty in value propagation through the transition model.
>    - Step 2: Bounding the immediate regret term presents two unique challenges as outlined in Section 4. Namely, (i) the target function $Q^\star_h(s, a) = f_R(s, a) + V^\star_{h+1}(f_S(s, a))$ is recursive and compositional, and thus cannot be directly modeled as a GP; and (ii) the TS algorithm samples from the posterior of a proxy $Q_{h,k}$, rather than from the true posterior over $Q^\star_h$. To address these issues, we derive high-probability upper and lower confidence bounds on the proxy and target value functions (Theorem 1). These bounds allow us to relate the sampled proxy values to the true values and quantify the regret incurred. This confidence analysis, which handles recursive composition under GP priors, is novel to our setting.
>    - Step 3: We extend the confidence bounds to the multi-step setting to analyze the propagation of uncertainty over the full episode.
>    - Step 4: Finally, we bound the uncertainty accumulated across all steps and episodes, which requires controlling the sum of posterior standard deviations over the sequence of inputs. In the multi-output setting, a naive approach would apply the standard elliptical potential lemma independently to each output dimension, resulting in a regret bound that scales with $\sqrt{d \Gamma(T)}$ where $d$ is the number of outputs. Instead, we introduce a new multi-output elliptical potential lemma (Lemma 1) that tracks uncertainty jointly across all outputs using the full output covariance structure. This avoids summing over independent bounds for each output and leads to a strictly tighter regret bound that scales with $\sqrt{\Gamma(T)}$ and is independent of the output dimension $d$. This improvement is essential for scalability and highlights the benefit of using a multi-output GP model rather than independent GPs.
>
>    In summary, while the algorithm we consider follows the general structure of posterior sampling, our analysis addresses a much more complex setting and introduces new analytical tools that are broadly applicable to related problems in RL and Bayesian learning (as noted by the reviewer). We therefore respectfully disagree with the reviewer’s assessment of the significance of our contribution as we believe that our work addresses an important and technically challenging problem of interest to the NeurIPS community.
>
> 2. *Lack of large-scale experiments.* \
> We would like to highlight that, as mentioned in the introduction, one of the motivations for our work is the recent empirical success of a closely related algorithm called HOT-GP that is studied by Bayrooti et al. (2025). HOT-GP achieved strong performance on standard benchmarks such as MuJoCo and VMAS and outperformed both model-free (e.g., PPO, SAC) and model-based (e.g., MBPO, UCB-based) methods. While their work demonstrated the algorithm’s practical utility, it did not provide theoretical guarantees. Our goal is complementary: we focus on establishing theoretical bounds for this class of algorithms by deriving the first sublinear regret bounds guaranteeing convergence to the performance of the optimal policy. To support our analysis, we include controlled numerical experiments that align closely with our theoretical framework and are designed to corroborate the analytical findings. Specifically, our experiments confirm sublinear regret and illustrate how the kernel complexity empirically influences the learning efficiency relative to the theoretical dependence on $\Gamma(\cdot)$.
>
> 3. *Assumptions.* \
> Below we clarify how Assumptions 1 and 2 relate to theory and practice.
>    - Assumption 1: GP models are often used for modeling complex dynamics in RL, particularly in low-data regimes where uncertainty quantification is critical. Notably, GP-based methods such as PILCO (Deisenroth and Rasmussen, 2011) and its successors have demonstrated strong empirical performance in continuous-control robotics. GPs offer flexible representation capacity as the smoothness of functions depends on the GP kernel choice. For example, the Mat&eacute;rn kernel family provides a smoothness parameter $\nu$ that controls function regularity. As shown in Srinivas et al. (2010), functions sampled from Mat&eacute;rn kernels can uniformly approximate any continuous function on compact subsets of $\mathbb{R}^d$, making the GP prior highly expressive. Lower values of $\nu$ yield rougher functions that better capture non-smooth behavior. Thus, this assumption holds for a broad class of reward and transition functions, including those with limited smoothness, so we believe it is not overly restrictive.
>    - Assumption 2: This is a mild technical condition required for Theorem 1 to ensure that value functions admit Taylor expansions with controlled error. This holds when the reward and transition functions are differentiable with bounded first and second derivatives.
>    - Assumptions 1 and 2 hold in many continuous-control tasks common in RL, such as the Gymnasium Pendulum and MountainCarContinuous tasks. These environments have smooth, deterministic transition dynamics and reward functions, making GP modeling appropriate and ensuring the value functions are sufficiently smooth.
>
>    While these assumptions do not capture all real-world dynamics, our setting still reflects many realistic control problems and provides an important intermediate step toward more general regret bounds. Moreover, our assumptions are relatively relaxed compared to standard assumptions in prior RL theory. For instance, the regret analysis for posterior sampling for RL (Osband et al., 2013; Osband et al., 2017) assumes a finite state-action space and does not extend to continuous MDPs, while RKHS-based regret analyses (e.g., Chowdhury & Gopalan, 2019) rely on predefined feature maps and a fixed kernel class, and linear MDP approaches (e.g., Jin et al., 2020) assume linearity in both rewards and transitions. In contrast, we give a complete analysis under one of the more general settings considered in the literature.
>
> Thanks again for your review and we would be happy to further discuss any of these points during the author-reviewer discussion period.

---

> > ### Comment · Reviewer_7dz9 · 2025-08-02
> >
> > We thank the authors for their careful and thoughtful response. I agree that the analyses in Step 2 and Step 4 represent technically meaningful and nontrivial contributions to the literature on Thompson Sampling with Gaussian Processes. That said, the current presentation blurs the line between genuinely novel components and parts that are, in essence, straightforward compositions of existing tools. For example, Theorem 1 appears to be a direct application of a Taylor expansion combined with standard GP confidence bounds.
> >
> > Regarding the empirical evaluation: while HOT-GP by Bayrooti et al. (2025) demonstrates strong performance on standard benchmarks, the current manuscript lacks any direct comparison to other competitive methods. I strongly recommend including at least one additional experiment that benchmarks the proposed approach against relevant baselines (e.g., standard TS with GPs or HOT-GP itself) on a representative task. Such a comparison, even on a single benchmark with well-chosen metrics and ablations, would greatly strengthen the empirical case and help quantify the practical gains from the proposed technical contributions.
> >
> > I will raise my rating to 4 because it may, as mentioned by other reviewers, be a stepping stone to future bounds for new algorithms. But I still recommend authors to add discussions in their rebuttals to their paper to clarify the novelty and assumptions, and conduct more experiments.

---

> > > ### Author Response · Authors · 2025-08-05
> > >
> > > Thank you very much for the additional feedback and we are glad that the rebuttal helped improve your evaluation of the paper. We appreciate your recognition that Steps 2 and 4 represent technically meaningful contributions to the literature on Thompson Sampling with GPs, and we sincerely value your engagement with our work. These are indeed the nontrivial and novel steps that address the core challenges of applying TS in our setting, while building on and extending tools from the existing literature. Following your suggestion, we will revise the paper to more clearly highlight the novel aspects of our analysis. In particular, we will incorporate the key points discussed in the rebuttal into the contributions section and reiterate them where appropriate in the main text to better contextualize the core contributions.
> > >
> > > We will also revise the paper to clarify the practicality of our assumptions. Specifically, we will emphasize that GP models are widely used in RL and that their flexibility depends on the choice of kernel. Additionally, we will expand the comparison with assumptions in prior work to better highlight the generality of our setting, as outlined in the rebuttal.
> > >
> > > Thank you again for your time, positive feedback, and constructive input, all of which will help improve the clarity and quality of the paper.

---

### Official Review · Reviewer_z43g · 2025-07-02

**Clarity:** 3
**Significance:** 3
**Originality:** 3
**Rating:** 5
**Confidence:** 3

**Summary:**

The authors propose performing posterior sampling in reinforcement learning (PSRL) for infinite state-action space using Gaussian processes over deterministic rewards and transitions. The authors established a sublinear regret bound, improving prior work that only provided Bayesian regret guarantees.

**Questions:**

- Could you explain how Assumption 1 was used in the proof?
- Is it possible to generalize the proof for the case of stochastic rewards and transitions (for example, in the Gaussian case)?
- Is it possible to compare your results and assumptions with the results and assumptions in the case of kernel RL (see, e.g., Vakili and Olkhovskaya 2023)? It seems that the dependence on H for your algorithm is better than presented in (Vakili and Olkhovskaya 2023), but I am not sure that it is possible to compare these settings.


Related work that would be good to discuss:
Fan, Y., Chen, L., & Wang, Y. (2018). Efficient model-free reinforcement learning using gaussian process. arXiv preprint arXiv:1812.04359.

Vakili, S., & Olkhovskaya, J. (2023). Kernelized reinforcement learning with order optimal regret bounds. Advances in Neural Information Processing Systems, 36, 4225-4247.

**Ethical Concerns:**

["NO or VERY MINOR ethics concerns only"]

**Final Justification:**

The paper presents a very elegant analysis in the Bayes-RL paradigm. After the rebuttal, the practical motivation of the method became clearer to me, and I happily increased my score to Accept.

**Limitations:**

The paper is of a theoretical nature and thus does not have a clear potential negative societal impact that needs to be discussed.

Regarding the theoretical limitation, the paper needs to discuss Assumption 1 in more detail.

**Paper Formatting Concerns:**

No formatting concerns.

**Quality:**

3

**Strengths And Weaknesses:**

## Strengths
- First worst-case regret bound for PSRL-type procedure with Gaussian process regression with a very short and elegant proof technique.
- Improvements from Bayesian to worst-case regret bound (although it is not discussed in the paper since it misses a reference to Fan et al. 2018. )

## Weaknesses
- Restrictive assumptions on deterministic rewards and transitions;
- Lack of large-scale experiments (on MuJoCo, for example) that may add additional value.

Fan, Y., Chen, L., & Wang, Y. (2018). Efficient model-free reinforcement learning using gaussian process. arXiv preprint arXiv:1812.04359.

---

> ### Author Rebuttal · Authors · 2025-07-31
>
> Thank you very much for reviewing our paper and for your positive and constructive feedback. Below we respond to your comments and questions in detail and we hope this will help improve your evaluation of the work.
>
> **Questions**
>
> 1. *How was Assumption 1 used in the proof?* \
> Assumption 1 is used in the proof to derive the confidence bounds that serve as a key building block in the regret analysis and are specifically used in Step 2 of the proof of Theorem 2 to bound the immediate regret. To elaborate more on this, our analysis is decomposed into four steps and Step 2 bounds the immediate regret, i.e., the loss in the optimal Q-function $Q^{\star}(s_h, a^{\star}_h) - Q^{\star}(s_h, a_h)$, where $a_h$ is the action selected by the Thompson sampling algorithm at step $h$. This step relies on Corollary 1, which in turn is based on Theorem 1. Briefly, Theorem 1 establishes high-probability confidence intervals for composed GP functions and Corollary 1 applies this to recursive value functions in RL. These confidence intervals crucially rely on Assumption 1, which models the joint reward and transition functions as a multi-output GP. To clarify this connection explicitly, we have added this sentence before introducing Theorem 1: “Under Assumption 1, the true reward and transition functions are modeled with a GP prior, allowing us to derive high-probability bounds for compositional functions of the form $v(f)$.” We hope this improves the clarity and would like to note that the full derivation can be found in Appendices A and B and is also outlined in the proof sketch in Section 4.
>
> 2. *Is it possible to generalize the proof for the case of stochastic rewards and transitions?* \
> Thank you for this insightful question. Generalizing to stochastic environments is indeed a natural next step and we view our work as a foundation for this direction. In the case of stochastic rewards, the extension is straightforward and does not affect our analysis. The confidence intervals remain valid as long as the GP noise parameter $\lambda^2$ is chosen to upper-bound the variance of the reward noise. This adjustment ensures the posterior remains well-calibrated and the regret analysis proceeds as in the deterministic case. In contrast, generalizing to stochastic transitions introduces new challenges as this setting corresponds to GP regression with input uncertainty, since the state $s$ that is used recursively in the value function would depend on a random variable. Addressing this would require nontrivial extensions beyond the standard GP framework used in our work. Developing such tools is outside the scope of this paper but we believe the techniques that we introduce provide a strong foundation for extending regret analysis to stochastic transition dynamics in future work.
>
> 3. *Comparing with kernel RL (Chowdhury and Gopalan, 2019; Yang and Wang, 2020; and Vakili and Olkhovskaya, 2023) and their dependence on $H$.*
> All of these works assume that the conditional probability distribution of the next state belongs to the RKHS of a known kernel: $p_h(\cdot \mid s,a) \in \mathcal{H}_k$ for all $(s,a)$ and each step $h$. In contrast, our assumption is that the function $f(\cdot \mid s,a)$, which maps state-action pairs to next states, can be modeled using a GP and this model remains fixed across all steps $h$ within an episode.
>
>    There are also key differences in algorithm design: prior works adopt UCB-based approaches while we analyze Thompson Sampling.
>
>    Our regret bounds are not only tighter by a factor of $H$, but also by a factor of $\sqrt{\Gamma(T)}$ compared to Chowdhury and Gopalan (2019) and Yang and Wang (2020). This is particularly important, as their $O(\Gamma(T)\sqrt{T})$ bounds can become vacuous (i.e., linear in $T$) for many practical kernels such as the Mat&eacute;rn family. A key reason is the confidence intervals used in our analysis, which are tighter due to the Bayesian modeling of transitions via $f$ rather than the non-Bayesian RKHS assumption on $p_h$.
>
>    Roughly speaking, in a non-formal comparison of the regret analyses, the improved confidence intervals contribute a $\sqrt{H}$ factor. In addition, while those works model $p_h(\cdot \mid s,a)$ separately at each step $h$ (i.e., non-homogeneous transitions), our model assumes that $f(\cdot \mid s,a)$ remains fixed across steps within an episode. This shared structure contributes another $\sqrt{H}$ improvement. We will highlight these distinctions in the related work section for better clarity.
>
>    The gap in confidence intervals under RKHS assumptions is also evident in the Bayesian optimization literature, and is formally stated in the following COLT open problem: Open Problem: Tight Online Confidence Intervals for RKHS Elements, Sattar Vakili, Jonathan Scarlett, Tara Javidi, COLT 2021.
>
> *Discussing Fan et al. (2018).* \
> Thank you for pointing out the related work by Fan et al. (2018). We will add the reference and include a comparison in the paper. Their analysis builds on a technique from Osband & Van Roy (2017), which was introduced as a general method for establishing Bayesian regret, that is, the expected regret averaged over the distribution of all problem instances. In contrast, our notion of regret is stronger: we establish a high-probability bound that holds uniformly over all instances of the problem. As a result, our analysis requires fundamentally different and novel proof techniques.
>
> *Lack of large scale experiments.* \
> We would like to highlight that, as mentioned in the introduction, one of the motivations for our work is the recent empirical success of a closely related algorithm called HOT-GP that is studied by Bayrooti et al. (2025). HOT-GP achieved strong performance on standard benchmarks such as MuJoCo and VMAS and outperformed both model-free (e.g., PPO, SAC) and model-based (e.g., MBPO, UCB-based) methods. While their work demonstrated the algorithm’s practical utility, it did not provide theoretical guarantees. Our goal is complementary: we focus on establishing theoretical bounds for this class of algorithms by deriving the first sublinear regret bounds guaranteeing convergence to the performance of the optimal policy. To support our analysis, we include controlled numerical experiments that align closely with our theoretical framework and are designed to corroborate the analytical findings. Specifically, our experiments confirm sublinear regret and illustrate how the kernel complexity empirically influences the learning efficiency relative to the theoretical dependence on $\Gamma(\cdot)$.
>
> Thanks again for your review and we would be happy to further discuss any of these points during the author-reviewer discussion period.

---

> > ### Comment · Reviewer_z43g · 2025-08-04
> >
> > I would like to thank the authors for the detailed answer! The position of this result in the literature becomes clearer to me, as well as a source of inspiration for the method. I am happy to increase my score to Accept.

---

> > > ### Author Response · Authors · 2025-08-05
> > >
> > > Thank you for acknowledging the rebuttal and for your recommendation for acceptance. We are glad the response helped clarify the motivation and positioning of our work. In the revision, we will incorporate key points from the rebuttal, including a clearer discussion of Assumption 1 and a more explicit comparison with Fan et al. (2018) and kernel RL methods, to better contextualize our contributions. Thank you again for your time and thoughtful feedback, which will help improve the overall clarity and quality of the paper.

---

### Official Review · Reviewer_19Xz · 2025-07-03

**Clarity:** 3
**Significance:** 3
**Originality:** 2
**Rating:** 4
**Confidence:** 4

**Summary:**

This paper presents a theoretical analysis of Thompson Sampling (TS) for finite-horizon episodic MDPs. The authors introduce an algorithm, RL-GPS, which uses a multi-output Gaussian Process (GP) to jointly model the environment's reward and transition functions. The core contribution is a formal "no-regret" bound, which proves that the algorithm's performance asymptotically converges to the optimal policy. The paper includes experiments in synthetic environments to validate the sublinear regret growth and explore the impact of different GP kernel choices.

**Questions:**

See weakness.

**Ethical Concerns:**

["NO or VERY MINOR ethics concerns only"]

**Final Justification:**

Authors have address my concern on the restrictions on the MDP setting and provide the literature about the empirical performance of a close method. This shows the significance of study theoretical property for the proposed method.

**Limitations:**

yes

**Paper Formatting Concerns:**

None.

**Quality:**

3

**Strengths And Weaknesses:**

## Strengths

- The paper provides a rigorous theoretical analysis for TS in a continuous-state RL setting.

- The paper is well-organized and clearly presents the problem, algorithm, and theoretical results.

## Weaknesses

- The reliance on a GP to model the environment dynamics is a major limitation. This assumption restricts the method's applicability to a very narrow class of problems where the true reward and transition functions are smooth and well-behaved. Many real-world episodic RL problems involve discontinuous, non-stationary dynamics that cannot be accurately captured by a standard GP model.

- The empirical evaluation is insufficient to demonstrate the practical value of the proposed algorithm. The experiments are limited to internal comparisons between different kernel choices for RL-GPS and do not include any established benchmark methods from model-based or model-free RL (e.g., UCB-based approaches, PPO, SAC).

---

> ### Author Rebuttal · Authors · 2025-07-31
>
> Thank you very much for taking the time to review our paper. We appreciate your comments and are glad that you find the paper to be clear and well-structured and the theoretical analysis to be rigorous. We address the weaknesses highlighted in detail below and hope our responses will help improve your evaluation of the paper.
>
> **Weaknesses**
>
> 1. *Relying on a GP to model the environment dynamics restricts the application to a narrow class of smooth, well-behaved true functions that may not be practical*. \
> We agree that GP modeling introduces an assumption about the true underlying environment that does not always hold (as explicitly stated in the Limitations section), however we believe it is not overly restrictive, especially when compared to existing regret analyses. To clarify this, we respond in several parts:
>    - GP models are often used for modeling complex dynamics in RL, particularly in low-data regimes where uncertainty quantification is critical. Notably, GP-based methods such as PILCO (Deisenroth and Rasmussen, 2011) and its successors have demonstrated strong empirical performance in continuous-control robotics.
>    - GPs offer flexible representation capacity as the smoothness of functions depends on the GP kernel choice. For example, the Mat&eacute;rn kernel family provides a smoothness parameter $\nu$ that controls function regularity. As shown in Srinivas et al. (2010), functions sampled from Mat&eacute;rn kernels can uniformly approximate any continuous function on compact subsets of $\mathbb{R}^d$, making the GP prior highly expressive. Lower values of $\nu$ yield rougher functions that better capture non-smooth behavior. Thus, our setting can model a broad class of reward and transition functions, including those with limited smoothness.
>    - Our assumptions are milder than many related works on regret bounds for RL. For instance, the regret analysis for posterior sampling for RL (Osband et al., 2013; Osband et al., 2017) assumes a finite state-action space and does not extend to continuous MDPs, while RKHS-based regret analyses (e.g., Chowdhury & Gopalan, 2019) rely on predefined feature maps and a fixed kernel class, and linear MDP approaches (e.g., Jin et al., 2020) assume linearity in both rewards and transitions. In contrast, our multi-output GP model flexibly models correlations between reward and transitions without relying on discretization, linearity, or fixed feature representations.
>    - Our assumption that the true joint reward and transition function is drawn from a multi-output GP enables a tractable regret analysis in continuous state-action spaces. While GPs may not capture all real-world dynamics, such as sharp discontinuities, this setting still reflects many realistic control problems (e.g., MuJoCo environments) and provides an important intermediate step toward more general regret bounds.
>
> 2. *Empirical evaluation is insufficient to demonstrate the practical value of the algorithm*. \
> We would like to highlight that, as mentioned in the introduction, one of the motivations for our work is the recent empirical success of a closely related algorithm called HOT-GP that is studied by Bayrooti et al. (2025). HOT-GP achieved strong performance on standard benchmarks such as MuJoCo and VMAS and outperformed both model-free (e.g., PPO, SAC) and model-based (e.g., MBPO, UCB-based) methods. While their work demonstrated the algorithm’s practical utility, it did not provide theoretical guarantees. Our goal is complementary: we focus on establishing theoretical bounds for this class of algorithms by deriving the first sublinear regret bounds guaranteeing convergence to the performance of the optimal policy. To support our analysis, we include controlled numerical experiments that align closely with our theoretical framework and are designed to corroborate the analytical findings. Specifically, our experiments confirm sublinear regret and illustrate how the kernel complexity empirically influences the learning efficiency relative to the theoretical dependence on $\Gamma(\cdot)$.
>
> Thanks again for your review and we would be happy to further discuss any of these points during the author-reviewer discussion period.

---

> > ### Comment · Reviewer_19Xz · 2025-08-06
> >
> > Thanks authors for the detailed defenses on the concerns I raised about the MDP setting and providing the complementary literature to show the empirical performance. I am happy to increase my score to borderline accept.

---

> > > ### Author Response · Authors · 2025-08-08
> > >
> > > Thank you for acknowledging the rebuttal and for your updated recommendation. We are glad that clarifying the generality of the model and discussing the empirical performance helped improve your evaluation of the work. Thank you again for your time and please let us know if there are any outstanding concerns or questions we can help resolve.

---

### Note · Authors · 2025-08-14

We thank the AC for facilitating the review process and the reviewers for their valuable feedback and engagement. We are glad they highlighted the paper’s clarity, technical soundness, and usefulness. Below is a summary of key review points and our clarifications.

**GP modelling:** GPs are a standard choice in low-data RL where calibrated uncertainty is important (e.g., PILCO). Their expressiveness comes from the kernel: for example, Matérn kernels can uniformly approximate any continuous function on compact subsets of $\mathbb{R}^d$ (Srinivas et al., 2010), with smoothness controlled by $\nu$—small $\nu$ capturing rougher dynamics. Our joint, multi-output GP models reward and transition together, capturing cross-correlations without discretisation or linearity, and covering a broad class of continuous-control tasks with smooth dynamics. This setting is more general than common finite-state or linear-MDPs while still lending itself to theoretical analysis.

**Assumptions:** We clarified how Assumption 1 is used in the analysis. Assumption 2 is a mild smoothness condition, weaker than in much prior work. These enable analysis while remaining practical for settings such as continuous-control tasks.

**Experiments:** Our experiments verify sublinear regret and kernel-complexity dependence consistent with our analysis. We will add theoretical reference curves for better clarity. The related HOT-GP (Bayrooti et al., 2025) shows strong performance on MuJoCo and VMAS tasks against popular baselines; our work is complementary, providing the theoretical analysis.

**Significance of contribution and technical novelty:** Thompson sampling has been studied since the 1930s, yet the first proofs for $K$-armed bandits only appeared in the 2010s. Extending TS analysis to MDPs with continuous state-action spaces and minimal assumptions is substantially harder. We provide the first high-probability no-regret guarantee for TS in finite-horizon continuous MDPs under joint GP modelling of rewards and transitions. Our analysis tackles several challenges, including non-Gaussian, recursive value functions and the proxy–posterior mismatch introduced by Bellman composition, which makes the problem much harder than TS in the special case of Bayesian Optimization; and new bounds on multi-output uncertainty with delayed updates, leading to improved overall regret bounds in both $K$ and $H$.



We again thank the AC and reviewers and hope this summary supports the decision process.

---

### Decision · Program_Chairs · 2025-09-17

**Decision:**

Accept (poster)

**Comment:**

The paper analyzes Thompson Sampling for episodic finite-horizon MDPs under a joint Gaussian-process model of rewards and transitions and proves a high-probability sublinear regret guarantee. The analysis tackles the non-Gaussian, recursive Bellman composition of value functions and introduces a multi-output elliptical potential lemma to control uncertainty jointly across outputs, yielding tighter bounds than applying single-output results independently. Controlled experiments are aligned to the theory and illustrate the predicted dependence on kernel complexity. The writing is clear and the technical development is careful.

Across the reviews and discussion, the overall evaluation is positive: the theoretical contribution is regarded as rigorous and well organized, with the empirical section serving primarily to corroborate the analysis. Remaining questions centered on the scope of GP assumptions and on clarifying what is genuinely new in the proof pipeline; the authors’ responses clarified assumptions, positioning, and how the key steps extend prior TS analyses, and they committed to adding reference curves and sharpening novelty statements in the camera-ready.

After weighing the overall feedback, the AC places particular weight on a detailed technical assessment that viewed the modular proof steps as meaningful contributions while asking for a clearer separation of new versus standard ingredients and a minimal comparative benchmark to better connect the theory to practice.
Overall, this is a nice paper with a solid result, clear exposition, and reusable tools. The AC recommends acceptance, with a camera-ready that more explicitly highlights what is new in the proof pipeline, clarifies the role of the assumptions, and, if space permits, please incorporate suggestions that arise from the discussions.